# Unsupervised Polychromatic Neural Representation for CT Metal Artifact Reduction

**Qing Wu**♣    **Lixuan Chen**♣    **Ce Wang**◇    **Hongjiang Wei**♡
**S. Kevin Zhou**♠    **Jingyi Yu**♣    **Yuyao Zhang**♣,*

♣ShanghaiTech University
♡Shanghai Jiao Tong University
♠University of Science and Technology of China
◇Institute of Computing Technology, Chinese Academy of Sciences
{wuqing, chenlx1, yujingyi, zhangyy8}@shanghaitech.edu.cn
wangce@ict.ac.cn   hongjiang.wei@sjtu.edu.cn   skevinzhou@ustc.edu.cn

## Abstract

Emerging neural reconstruction techniques based on tomography (*e.g.*, NeRF, NeAT, and NeRP) have started showing unique capabilities in medical imaging. In this work, we present a novel **Poly**chromatic **ne**ural **r**epresentation (**Polyner**) to tackle the challenging problem of CT imaging when metallic implants exist within the human body. CT metal artifacts arise from the drastic variation of metal's attenuation coefficients at various energy levels of the X-ray spectrum, leading to a nonlinear metal effect in CT measurements. Recovering CT images from metal-affected measurements hence poses a complicated nonlinear inverse problem where empirical models adopted in previous metal artifact reduction (MAR) approaches lead to signal loss and strongly aliased reconstructions. Polyner instead models the MAR problem from a nonlinear inverse problem perspective. Specifically, we first derive a polychromatic forward model to accurately simulate the nonlinear CT acquisition process. Then, we incorporate our forward model into the implicit neural representation to accomplish reconstruction. Lastly, we adopt a regularizer to preserve the physical properties of the CT images across different energy levels while effectively constraining the solution space. Our Polyner is an unsupervised method and does not require any external training data. Experimenting with multiple datasets shows that our Polyner achieves comparable or better performance than supervised methods on in-domain datasets while demonstrating significant performance improvements on out-of-domain datasets. To the best of our knowledge, our Polyner is the first unsupervised MAR method that outperforms its supervised counterparts. The code for this work is available at: https://github.com/iwuqing/Polyner.

## 1   Introduction

Computed tomography (CT) is a widely-used imaging technique for clinical diagnosis. CT measurements are produced by passing X-rays at different angles through the human body. Reconstruction of CT images from these measurements is often approximately modeled as a linear inverse problem [1]. However, the attenuation coefficients of the metal materials dramatically vary with the X-ray spectral energy, which largely differs from those of human body tissues. This results in a non-negligible nonlinear metal effect in the CT measurements when the body carries metallic implants [2]. Hence,

---

*Corresponding author.

37th Conference on Neural Information Processing Systems (NeurIPS 2023).

recovering CT images from metal-affect measurements is a complicated *nonlinear inverse problem*. However, linear inverse problem algorithms, *e.g.*, filtered back projection (FBP) [3], always lead to severe artifacts in the resulting CT images. These artifacts can greatly hamper the accuracy of clinical diagnosis and treatment planning.

Many efforts have been made for CT metal artifact reduction (MAR) task [4]. Most studies [5, 6, 7, 8, 9, 10, 11, 12] simplify this task as a linear inverse problem by designing various schemes to replace metal-affected extreme value signals (*i.e.*, metal traces) in the measurements, which unfortunately results in significant degradation of the CT reconstruction due to the signal loss. Currently, supervised deep learning (DL) approaches [7, 8, 13, 14, 15, 16, 17, 6] are the mainstream MAR solutions. Supervised methods typically tend to learn mappings from metal-corrupted measurements to artifact-free CT images by training deep neural networks on a large-scale external dataset. However, related works face two main limitations. First, collecting numerous paired metal-corrupted measurements and high-quality CT images for training is resource-intensive and laborious. Second, if the metal shapes and the CT acquisition settings differ from those in the training dataset, performance may significantly decline. These challenges significantly limit the practical application of supervised MAR methods in clinical scenarios.

Implicit neural representation (INR) is a new paradigm to solve inverse problems [18]. It uses a multi-layer perception (MLP) to model an underlying signal by mapping coordinates to signal responses. The learning bias of the MLPs towards low-frequency signal components [19, 20] is considered as an implicit regularization term for the inverse problems. By approximately simulating the CT acquisition forward model as a linear integral, INR has achieved significant progress in the linear sparse-view CT reconstruction [21, 22, 23, 24, 25, 26]. However, its potential for tackling the nonlinear inverse problem, such as the MAR task, remains unexplored.

In this work, we aim to address the issue of MAR in CT scans from a nonlinear perspective. This allows us to use the complete measurement signals (*i.e.*, including metal traces) and thus achieve better CT reconstruction performance. Specifically, we model the MAR as a polychromatic (*i.e.*, multiple X-ray spectral energy levels) reconstruction task of CT image. This task is a highly ill-posed nonlinear inverse problem. To address this, we propose **Poly**chromatic **ne**ural **r**epresentation (**Polyner**), a novel extension of INR for the nonlinear problems. In our Polyner, we make three key designs. First, we derive a forward model that accurately simulates the polychromatic nonlinear CT acquisition process. Second, we define a regularization term that effectively constrains the solution space by preserving the physical properties of the CT images across different spectral energy levels. Third, we incorporate our forward model into the INR, granting its capacity to solve the nonlinear inverse problem. Meanwhile, the continuous prior brought by INR also contributes to a more stable solution. Our Polyner is an unsupervised method and does not require any external training data, which makes it potentially applicable to a wide range of CT imaging scenarios.

We evaluate the performance of our Polyner on three datasets, including two simulation datasets and one real-world collected dataset. The experimental results demonstrate that our Polyner performs comparably to the supervised DL method on in-domain datasets, while significantly outperforming them on out-of-domain datasets. We also conduct extensive ablation studies for several key designs of our method, verifying their effectiveness. *To the best of our knowledge, our Polyner is the first unsupervised MAR method superior to its supervised counterparts.*

The main contributions of this work can be summarised as follows: (1) We propose Polyner, an unsupervised polychromatic neural representation, that can recover high-quality CT images from metal-corrupted measurements without the need for any extra data. (2) We introduce a new model perspective for the CT MAR problem, *i.e.*, recovering polychromatic CT images to mitigate the nonlinear metal effect and thus achieving effective MAR. (3) We technically incorporate a polychromatic CT forward model into the INR, enabling the reconstruction of polychromatic CT images.

## 2   Preliminaries & Related Work

In this section, we first theoretically present the nonlinear metal effect during the CT acquisition process (Sec. §2.1). Then, we provide a brief overview of the literature on the metal artifacts reduction approaches (Sec. §2.2) and implicit neural representations for CT reconstruction (Sec. §2.3).

## 2.1 Nonlinear Metal Effect in CT Measurements

The physical interaction of X-rays passing through an object is governed by Lambert-Beer's law [27, 28], which can be expressed as follows:

$$I(\mathbf{r}, E_0) = I_0(E_0) \cdot \exp\left(-\int_{\mathbf{r}} \mu_{E_0}(\mathbf{x}) \mathrm{d}\mathbf{x}\right), \tag{1}$$

where $\mathbf{r}$ denotes a monochromatic X-ray, $E_0$ is its energy level, $I_0(E_0)$ is the number of photons emitted by an X-ray source, $I(\mathbf{r}, E_0)$ is the number of photons received by an X-ray detector, and $\mu_{E_0}(\mathbf{x})$ is the linear attenuation coefficient (LAC) of the observed object at the position $\mathbf{x}$. The LACs serve as a measure of the object's ability to absorb the X-rays, with higher values indicating greater absorption ability.

In practice, X-rays often are polychromatic due to limitations in the X-ray source [2, 29]. Consider a polychromatic X-ray composed of multiple monochromatic X-rays belonging to an energy level range $\mathcal{E}$, by combining Eq. (1), the CT measurement data $p(\mathbf{r})$ thus can be written as below:

$$\begin{aligned} p(\mathbf{r}) &= -\ln \frac{\int_{\mathcal{E}} I(\mathbf{r}, E) \mathrm{d}E}{\int_{\mathcal{E}} I_0(E) \mathrm{d}E} \\ &= -\ln \int_{\mathcal{E}} \eta(E) \cdot \exp\left(-\int_{\mathbf{r}} \mu_E(\mathbf{x}) \mathrm{d}\mathbf{x}\right) \mathrm{d}E, \quad \text{with } \eta(E) = \frac{I_0(E)}{\int_{\mathcal{E}} I_0(E') \mathrm{d}E'}, \end{aligned} \tag{2}$$

where $\eta(E)$ is the normalized energy spectrum that characterizes the distribution of the number of photons $I_0(E)$ emitted by the X-ray source over the energy level $E$.

CT reconstruction refers to solve the LACs $\mu$ from the measurement data $p$. It is widely recognized that the LACs of human body tissues exhibit slow decreases with increasing the X-ray's energy level, while those of metals undergo sharp variations as a function of energy level [2, 7, 8, 30]. Therefore, a usual assumption for the human body is that $|\mu_{E_a}(\mathbf{x}) - \mu_{E_b}(\mathbf{x})| \approx 0, \ \forall E_a, E_b \in \mathcal{E}$, i.e., the LACs of the human body are considered as energy-independent.

Based on this assumption, we can represent the underlying LACs of a human body with metallic implants as follows:

$$\mu_E(\mathbf{x}) = \mathcal{M}(\mathbf{x}) \cdot \mu_E^\star(\mathbf{x}) + [1 - \mathcal{M}(\mathbf{x})] \cdot \mu(\mathbf{x}), \tag{3}$$

where $\mu_E^\star(\mathbf{x})$ and $\mu(\mathbf{x})$ denote the LACs of the metal and human body, respectively. $\mathcal{M}$ is a metal binary mask ($\mathcal{M} = 1$ for the metal region and $\mathcal{M} = 0$ otherwise).

By combining Eq. (2), its CT measurement data can be derived as:

$$\begin{aligned} p(\mathbf{r}) &= -\ln \int_{\mathcal{E}} \eta(E) \cdot \exp\left(-\int_{\mathbf{r}} \mathcal{M}(\mathbf{x}) \cdot \mu_E^\star(\mathbf{x}) + [1 - \mathcal{M}(\mathbf{x})] \cdot \mu(\mathbf{x}) \mathrm{d}\mathbf{x}\right) \mathrm{d}E \\ &= \underbrace{-\ln \int_{\mathcal{E}} \eta(E) \cdot \exp\left(-\int_{\mathbf{r}} \mathcal{M}(\mathbf{x}) \cdot \mu_E^\star(\mathbf{x}) \mathrm{d}\mathbf{x}\right) \mathrm{d}E}_{\text{Nonlinear Metal Effect}} + \underbrace{\int_{\mathbf{r}} [1 - \mathcal{M}(\mathbf{x})] \cdot \mu(\mathbf{x}) \mathrm{d}\mathbf{x}}_{\text{Linear Integral}}. \end{aligned} \tag{4}$$

This implies that the CT measurement of a human body with metallic implants can be divided into two parts: (1) a linear integral transformation for the human body and (2) a nonlinear transformation for the metallic implants. The latter is also known as the *nonlinear metal effect* in CT measurements. Therefore, using standard reconstruction techniques, such as FBP [3], which are designed for linear inverse problems, will result in severe metal artifacts in the reconstructed CT image.

## 2.2 CT Metal Artifact Reduction Approaches

Classical model-based algorithms [10, 11, 12] formulate the MAR as an image inpainting task. The metal traces in the measurements are treated as missing and completed by interpolation. However, these model-based methods often produce severe secondary artifacts since the interpolation algorithms ignore the CT physical geometry constraints. With the great success of deep neural networks in low-level vision tasks [31, 32], numerous supervised DL methods have been proposed [7, 8, 13, 14, 6, 15, 17, 16]. For example, Lin *et al.* [8] proposed a dual domain network (DuDoNet) to

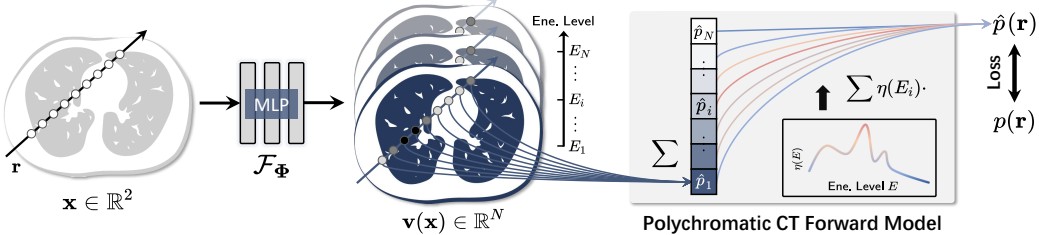

Figure 1: Overview of the proposed Polyner model. Firstly, we sample coordinates $\mathbf{x}$ along an X-ray $\mathbf{r}$. Then, we feed these coordinates into an MLP to predict the corresponding LACs $\mathbf{v}(\mathbf{x}) = \{\mu_1(\mathbf{x}), \cdots, \mu_N(\mathbf{x})\}$ at $N$ energy levels. Thirdly, we leverage a differentiable polychromatic CT forward model to transform the predicted LACs into CT measurements $\hat{p}(\mathbf{r})$. Finally, we optimize the MLP by minimizing the loss between the predicted and real CT measurements.

simultaneously complete the measurements and enhance the CT images. Wang *et al*. [13] presented an adaptive convolutional dictionary network (ACDNet), incorporating prior knowledge of metal artifacts into a deep network. These supervised methods achieve state-of-the-art (SOTA) MAR performance due to their well-designed network architectures and data-driven priors. However, they require a large-scale training dataset for supervised learning and often suffer from out-of-domain (OOD) problems [9, 30], severely limiting their usefulness in real-world scenarios. In contrast, our Polyner is a fully unsupervised method that does not require any external training data. Moreover, both model-based and supervised DL methods often treat metal traces as missing data, which severely hinders the restoration of image details near metals. Our method decomposes the CT measurement with polychromatic LACs at different energy levels as in Eq. (2), and directly utilizes the complete measurement as input data, which avoids signal loss of the metal traces and thus achieves a better MAR performance.

### 2.3 Implicit Neural Representation for CT Reconstruction

Implicit neural representation (INR) is a novel paradigm to continuously parameterize a variety of signals. INR represents an underlying signal as a continuous function that maps spatial positions to signal responses and uses an MLP network to approximate the complex function. Due to the learning bias of the MLP networks to low-frequency signals [19, 20], INR can be used for various vision inverse problems, *e.g.*, view synthesis [33, 34], and surface reconstruction [35, 36]. Recently, many INR-based approaches [22, 23, 24, 25, 37, 21, 38, 39, 40, 26, 41] have been emerged for CT reconstruction. There are two key designs in these methods: (1) representing the unsolved object as a function that maps spatial coordinates to its corresponding monochromatic LCAs at a single energy level, and (2) combining line integral transformation with an MLP network to approximate the function. They have shown excellent CT reconstruction performance benefiting from implicit priors by INR. However, these methods fail to handle the nonlinear CT MAR task because they strictly suppose the unsolved LACs are monochromatic, *i.e.*, energy-independent, leading to the nonlinear metal effect being ignored. In contrast, our method leverages a polychromatic forward model to accurately describe the formation of the metal effect and thus enables an effective MAR.

## 3 Proposed Method

In this section, we propose our Polyner model. We first introduce our modeling for the CT MAR problem (Sec. §3.1). Next, we present a polychromatic CT forward model for simulating the CT acquisition process (Sec. §3.2). Moreover, we define the loss function to optimize our Polyner (Sec. §3.3). Finally, we provide the pipeline to recover the reconstruction results (Sec. §3.4). An overview of the proposed Polyner model is shown in Fig. 1.

### 3.1 Problem Formulation

The LACs of metals to X-rays vary significantly with the spectral energy level of the X-ray, resulting in metal-affect extreme signal values in the measurement data (*i.e.*, nonlinear metal effect discussed in

Sec. §2.1). Recovering the underlying CT image from such measurement is a complicated nonlinear inverse problem. Existing MAR approaches [5, 6, 7, 8, 9, 10, 11, 12] mostly formulated it as a linear inverse problem by removing the metal-affected parts in the measurement.

Instead, we aim to address the MAR problem from a nonlinear perspective. We suppose that the polychromatic X-rays in the CT acquisition can be decomposed into monochromatic X-rays at $N$ discrete energy levels. Then, we propose to reconstruct the polychromatic CT images at each of the $N$ energy levels (*i.e.*, a total number of $N$ LAC-maps at each monochromatic X-ray for the observed object). This reconstruction task is nonlinear and is independent to the nonlinear metal effect. Hence, the MAR task is modeled as a reconstruction problem of the polychromatic CT images.

To accomplish the reconstruction, we represent the underlying object as a continuous function of spatial coordinate, which can expressed as below:

$$f : \mathbf{x} = (x, y) \in \mathbb{R}^2 \longrightarrow \mathbf{v}(\mathbf{x}) = \{\mu_1(\mathbf{x}), \cdots, \mu_i(\mathbf{x}), \cdots, \mu_N(\mathbf{x})\} \in \mathbb{R}^N, \tag{5}$$

where $\mathbf{x}$ denotes any spatial coordinate and $\mathbf{v}(\mathbf{x})$ is the polychromatic CT image value at that position. Here the item $\mu_i(\mathbf{x})$ is the LAC of the object to the monochromatic X-ray at energy level $E_i$.

However, the function $f$ is very complicated and intractable. Hence, we leverage an MLP network $\mathcal{F}_{\mathbf{\Phi}}$ (here $\mathbf{\Phi}$ denotes trainable weights) to approximate it. In other words, we learn a neural representation of the function $f$. The artifacts-free CT images can be reconstructed by feeding all the spatial coordinates into the well-optimized MLP network $\mathcal{F}_{\mathbf{\Phi}}$.

## 3.2 Polychromatic CT Forward Model

To learn the function $f$, we optimize the MLP network's trainable weights $\mathbf{\Phi}$ to map any spatial coordinate $\mathbf{x}$ in space into its corresponding polychromatic LACs $\mathbf{v}(\mathbf{x})$. Therefore, a *differentiable* forward model is required for transforming the LACs predicted by the MLP network $\mathcal{F}_{\mathbf{\Phi}}$ into the CT measurements while also allowing the back-propagation of gradients from the measurement domain to the image domain.

We derive a polychromatic forward model for X-ray CT systems in Eq. (2). This forward model can accurately describe the complicated nonlinear CT acquisition process. Our Polyner employs its discrete form to transform the MLP-predicted polychromatic LACs $\mathbf{v}(\mathbf{x})$ into the measurement data $\hat{p}$. Fig. 1 illustrates its pipeline. For the LACs $\mu_i$ at the energy level $E_i$, we first generate its projection value $\hat{p}_i(\mathbf{r})$ along the X-ray $\mathbf{r}$ using a linear summation operator. We then leverage the normalized energy spectrum $\eta(E_i)$ to weight the sum of $\hat{p}_i(\mathbf{r})$ and produce the final measurement data $\hat{p}(\mathbf{r})$. Formally, the polychromatic CT forward model can be expressed as follows:

$$\hat{p}(\mathbf{r}) = -\ln \sum_{i=1}^{N} \eta(E_i) \cdot \exp\{-\hat{p}_i(\mathbf{r})\}, \quad \text{with } \hat{p}_i(\mathbf{r}) = \sum_{\mathbf{x} \in \mathbf{r}} \mu_i(\mathbf{x}) \cdot \Delta\mathbf{x}, \tag{6}$$

where $\Delta\mathbf{x}$ represents the distance between adjacent sampled coordinates along the X-ray $\mathbf{r}$. It is set as the physical resolution (*i.e.*, voxel size) of the reconstructed CT image. Here, the normalized energy spectrum $\eta \in \mathbb{R}^N$ is considered as a known prior knowledge. In our experiments, we leverage SPEKTR toolkit developed by Punnoose *et al.* [42] to estimate it.

## 3.3 Loss Function

**Data Consistency Loss.** We first compute a data consistency (DC) loss that measures the distance between the predicted and measured measurements. The DC loss is implemented by $\ell_1$ norm, which can be written as below:

$$\mathcal{L}_{\text{DC}} = \frac{1}{|\mathcal{R}|} \sum_{\mathbf{r} \in \mathcal{R}} \|p(\mathbf{r}) - \hat{p}(\mathbf{r})\|_1, \tag{7}$$

where $\mathcal{R}$ denotes a set of the sampling X-rays $\mathbf{r}$ at each training iteration.

**Energy-dependent Smooth Loss.** As discussed in Sec. §2.1, the LACs of the human body tissues to X-rays exhibit smooth changes with increasing the energy level of the X-rays [7, 8, 30]. We thus propose an energy-dependent smooth (EDS) loss to preserve this physical property. Specifically, for the human body, the proposed EDS loss enforces a regularization that constrains the changes between the LACs at any adjacent energy levels to be smooth. We implement this regularization using the $\ell_1$

norm of the gradient along the energy spectrum. It is approximated by the sum of the absolute error between the predicted LACs at any adjacent X-ray energy levels. Mathematically, the EDS loss can be written as follows:

$$\mathcal{L}_{\text{EDS}} = \frac{1}{|\mathcal{R}| \cdot |\mathbf{r}|} \sum_{\mathbf{r} \in \mathcal{R}} \sum_{\mathbf{x} \in \mathbf{r}} [1 - \mathcal{M}(\mathbf{x})] \cdot \|\nabla_E \mathbf{v}(\mathbf{x})\|_1$$

$$= \frac{1}{|\mathcal{R}| \cdot |\mathbf{r}|} \sum_{\mathbf{r} \in \mathcal{R}} \sum_{\mathbf{x} \in \mathbf{r}} [1 - \mathcal{M}(\mathbf{x})] \cdot \sum_{i=2}^{N} |\mu_{i-1}(\mathbf{x}) - \mu_i(\mathbf{x})|,$$

(8)

where $\mathcal{M}$ is a metal binary mask ($\mathcal{M} = 1$ for the metal region and $\mathcal{M} = 0$ otherwise).

In summary, our total loss function is defined as below:

$$\mathcal{L} = \mathcal{L}_{\text{DC}} + \lambda \cdot \mathcal{L}_{\text{EDS}},$$

(9)

where $\lambda \geq 0$ is a hyper-parameter that controls the contribution of the EDS regularization $\mathcal{L}_{\text{EDS}}$.

### 3.4 Reconstruction of Artifact-free CT Image

Once the optimization is completed, the MLP network $\mathcal{F}_{\mathbf{\Phi}}$ is able to predict the polychromatic LACs $\mathbf{v}(\mathbf{x}) = \{\mu_1(\mathbf{x}), \cdots, \mu_i(\mathbf{x}), \cdots, \mu_N(\mathbf{x})\}$ at $N$ energy levels for any input spatial coordinate $\mathbf{x}$, meaning the polychromatic CT images can be reconstructed. Given a polychromatic X-ray that consists of multiple monochromatic X-rays at $N$ energy levels, its equivalent monochromatic X-ray's energy can be approximated by $\bar{E} = \frac{1}{N} \sum_{i=1}^{N} E_i$. In this work, we aim to solve the CT MAR problem. Thus, we employ the LACs $\mu_{\bar{E}}$ at the energy level $\bar{E}$ as our final reconstructed image.

## 4 Experiments

In this section, we aim to answer two key questions: (1) Can our unsupervised Polyner compete with supervised DL MAR techniques for the in-domain and out-of-domain datasets? (2) What is the impact of the key components in our Polyner on the model performance? To study these questions, we conduct comprehensive experiments on three datasets. We provide additional experiment details and results in the supplementary material.

### 4.1 Experimental Settings

**Datasets.** We conduct experiments on four datasets, including two simulation datasets and two real collected datasets. The first dataset is DeepLesion [43], the most commonly used dataset for the CT MAR evaluation. For this dataset, we extract 200 2D slices of 256×256 size from the original 3D CT volumes as ground truth (GT). Then, we follow the pipeline described in [30, 7, 8] to synthesize metal-corrupted sinograms as input data. The second dataset is the XCOM dataset [44], for which we employ two cases provided by Zhang et al. [7]. The third is a real-world dataset. As shown in Fig. 3 (*Left*), we insert a metal paper clip into a walnut sample and scan it with a commercial Bruker SKYSCAN 1276 micro-CT scanner. The fourth is also a real-world dataset. We use the same micro-CT scanner to scan a mouse tight containing a metal intramedullary needle. *Note that all data solely are utilized for testing purposes since our method is fully unsupervised.*

**Baselines & Metrics.** We employ nine representative CT MAR approaches from three categories as baselines: (1) four model-based algorithms (FBP [3], LI [10], NMAR [11], and ART [45]); (2) three supervised CNN-based DL methods (CNN-MAR [7], DICDNet [14], and ACDNet [13]); and (3) two unsupervised DL method (ADN [30] and Score-MAR [9]). *To ensure a fair comparison, we evaluate the five DL models using the checkpoints provided by the authors.* Among them, CNN-MAR [7] and Score-MAR [9] are respectively trained on the XCOM [44] and LIDC [46] datasets, while the other three methods are trained on the DeepLesion dataset [43]. We use peak signal-to-noise ratio (PSNR) and structural similarity index measure (SSIM) to quantitatively evaluate the MAR performance.

**Implementation Details.** For our Polyner, we leverage hash encoding [47] in combination with two fully connected (FC) layers of width 128 to implement the MLP network. A ReLU activation function is then applied after the first FC layer. For the hash encoding [47], we configure its hyper-parameters as follow: $L = 16$, $T = 2^{19}$, $F = 8$, $N_{\min} = 2$, and $b = 2$. To optimize the network, we randomly

Table 1: Quantitative results of compared methods on the DeepLesion [43] and XCOM [44] datasets. The best and second performances are respectively highlighted in **bold** and underline.

| Category | Method | DeepLesion [43] | | XCOM [44] | |
|---|---|---|---|---|---|
| | | PSNR | SSIM | PSNR | SSIM |
| Model-based | FBP [3] | 29.98±3.46 | 0.7470±0.0992 | 23.76±0.08 | 0.6506±0.0240 |
| | LI [10] | 31.85±3.20 | 0.8483±0.0726 | 33.57±0.44 | 0.8866±0.0300 |
| | ART [45] | 32.88±3.63 | 0.8352±0.0701 | 32.45±3.76 | 0.8322±0.0762 |
| | NMAR [11] | 32.84±5.07 | 0.8920±0.0938 | 36.63±0.23 | 0.9473±0.0068 |
| Supervised | CNN-MAR [7] | 34.22±2.70 | 0.9240±0.0375 | 38.52±0.77 | **0.9591±0.0100** |
| | DICDNet [14] | 37.55±2.52 | 0.9689±0.0116 | 32.65±2.11 | 0.9449±0.0037 |
| | ACDNet [13] | **38.19±2.54** | 0.9675±0.0152 | 33.04±1.65 | 0.9343±0.0089 |
| Unsupervised | ADN [30] | 33.00±3.21 | 0.9412±0.0246 | 25.26±0.94 | 0.8737±0.0107 |
| | Score-MAR [9] | 31.66±3.72 | 0.9174±0.0423 | 24.92±1.15 | 0.8754±0.0044 |
| | Polyner (Ours) | 37.57±1.93 | **0.9754±0.0083** | **38.74±0.59** | 0.9508±0.0001 |

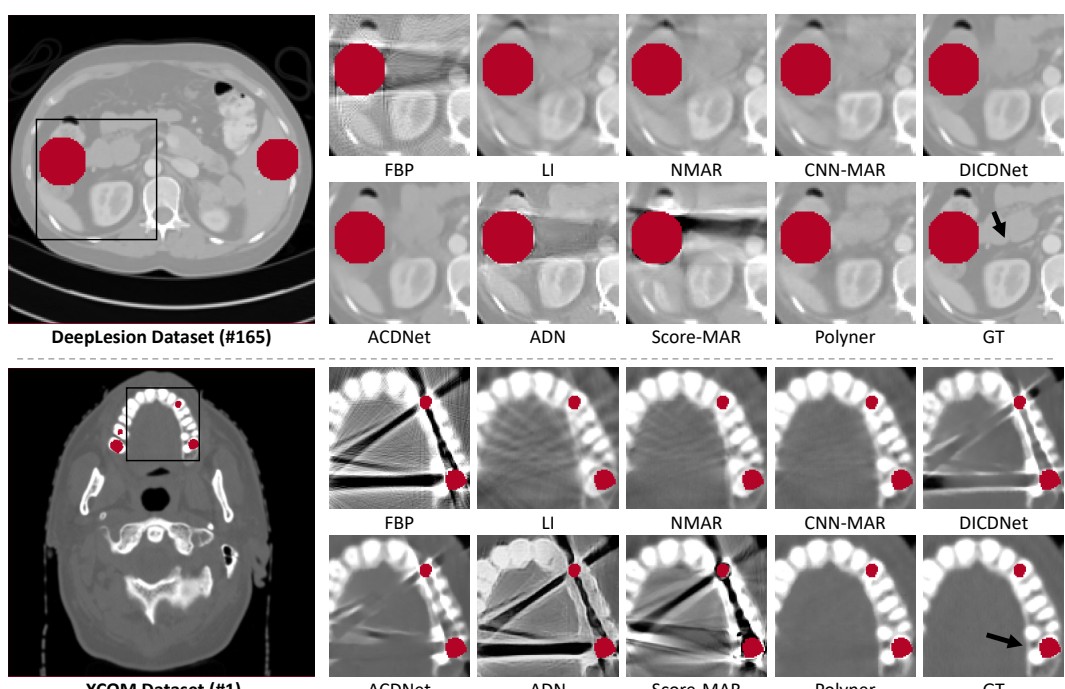

Figure 2: Qualitative results of the compared methods on two samples (#124 and #2) of the DeepLesion [43] and XCOM[44] datasets. The red regions denote metals.

sample 80 X-rays, *i.e.*, $|\mathcal{R}| = 80$ in Eqs. (7) & (8), at each training iteration. We set the hyperparameter $\lambda$ to 0.2 in Eq. (8). We employ the Adam optimizer [48] with default hyper-parameters and set the initial learning rate to 1e-3, which decays by a factor of 0.5 per 1000 epochs. The total number of training epochs is 4000. *It is worth noting that all hyper-parameters are tuned on 10 samples from the DeepLesion dataset [43], and are then held constant across all other samples.*

## 4.2 Main Results

**Comparison on Simulation Data.** Table 1 presents the quantitative results. On the DeepLesion dataset [43], ACDNet [13], our Polyner, and DICDNet [14] achieve the top three performances and significantly outperform other baseline models. Specifically, our Polyner produces the second-best performance, trailing slightly behind ACDNet [13] by only -0.62 dB in PSNR. However, on the XCOM dataset [44], we observe that DICDNet and ACDNet trained on the DeepLesion dataset [43] suffer from severe performance drops due to the OOD problem. Their performance is even lower than

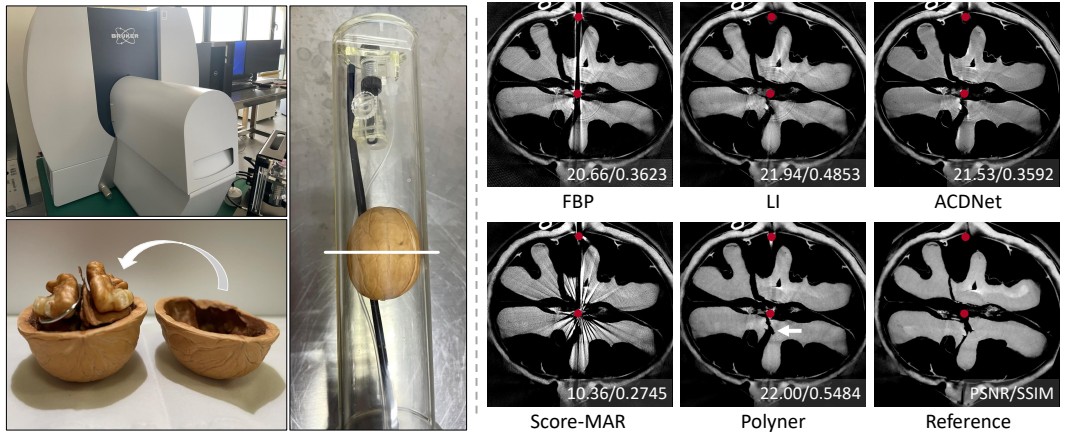

Figure 3: (*Left*) A Bruker SKYSCAN 1276 micro-CT scanner and a walnut sample with a metal paper clip. (*Right*) Qualitative results of the compared methods on the walnut sample data. The red regions denote metals.

that of the model-based NMAR [11] algorithm, with -3.98 dB and -3.59 dB in PSNR, respectively. In contrast, our Polyner obtains the best performance, slightly outperforming CNN-MAR trained on the XCOM dataset [44] by +0.22 dB in PSNR. Fig. 2 shows the qualitative results. Notably, three model-based algorithms (FBP [3], LI [10], and NMAR [11]) and two unsupervised methods (ADN [30] and Score-MAR [9]) cannot yield satisfactory results, exhibiting severe artifacts on both datasets. Conversely, while DICDNet [14] and ACDNet for the DeepLesion dataset [43] yield excellent MAR performances, their results on the XCOM dataset [44] include severe artifacts. This contrasts with CNN-MAR [7], which achieves superior results for the XCOM dataset [44] but not for the DeepLesion dataset [43]. These visual inspections are consistent with the above quantitative comparisons. Remarkably, our proposed Polyner presents clean and fine-detail CT reconstructions for both datasets, indicating its robustness and superiority over existing MAR methods. *More visual results can be found in the supplementary material.*

**Comparison on Real Data.** Though our Polyner model performs best on the two simulation datasets, it is important to study its performance on real collected CT data. To achieve this, we insert a metal paper clip into a walnut sample and then scan it with a commercial Bruker SKYSCAN 1276 micro-CT, as shown in Fig 3 (*Left*). We compare five baselines with our Polyner for reconstructing a 2D slice of the sample. The remaining three baselines (NMAR [11], CNN-MAR [7], and ADN [30]) are not compared because they require prior knowledge of water-bone segmentation, which is invalid for the walnut sample. Fig. 3 (*Right*) demonstrates the qualitative results. FBP [3] performs the worst, producing severe shadow artifacts. LI [10] exhibits local deformations in the reconstructed images. Score-MAR [9] almost fails due to the domain shift problem. ACDNet [13] is not effective in completely removing these shadow artifacts, it instead produces an offset in image contrast. In comparison, our Polyner recovers the best visual results in both image details and contrast. We also present the quantitative results in Fig. 3. The proposed Polyner achieves the best performance in two metrics. However, it is worth noting that the quantitative performances of all methods are relatively low (the best PSNR of 22 dB by our Polyner) since the reference CT image cannot be considered GT due to the non-negligible non-rigid deformation caused by the metal paper clip insertion. *In addition, we compare our Polyner with FDK algorithm [49] on the real mouse tight sample scanned by a 3D cone-beam CT geometry. Related results are provided in the supplementary material.*

### 4.3 Ablation Studies

**Influence of Polychromatic CT Forward Model.** We explore the efficiency of the polychromatic CT forward model in our Polyner. To this end, we replace it with the linear integral transformation as in [21, 22, 23, 24, 25, 26]. Other model configurations are kept the same for a fair comparison. Fig. 4 demonstrates the qualitative results. From the visualization, it is clear that without the forward model, our Polyner almost fails to handle the shadow artifacts caused by the nonlinear metal effect. In comparison, our full model with the forward model produces a clean CT image that is very close

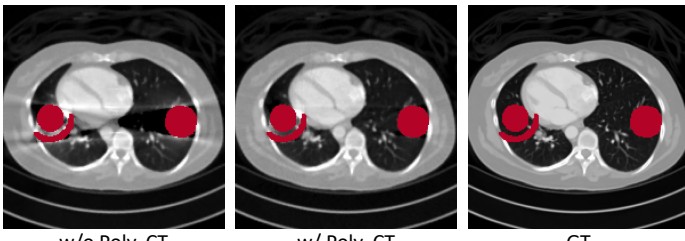

| | | |
| w/o Poly. CT | w/ Poly. CT | GT |

Figure 4: Qualitative results of our Polyner ablating the polychromatic CT forward model on a sample (#6) of the DeepLesion dataset [43]. The red regions denote metals.

Table 2: Quantitative results of our Polyner ablating the polychromatic CT forward model on the DeepLesion dataset [43].

| Module | PSNR |
| --- | --- |
| w/o Poly. CT | 32.95±4.31 |
| w/ Poly. CT | **36.87±1.56** |

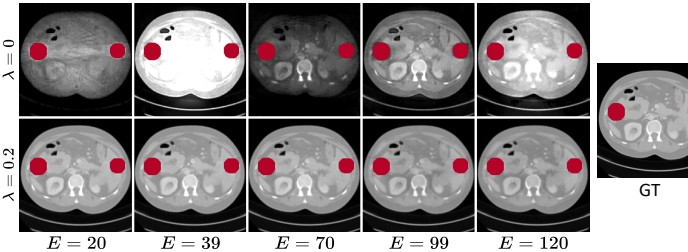

Figure 5: Qualitative comparison of the CT images at different energy levels using our Polyner model without (*Top Row*) and with (*Bottom Row*) the EDS loss $\mathcal{L}_{\text{EDS}}$ on a sample (#4) of the DeepLesion dataset [43]. The red regions denote metals.

Table 3: Quantitative results of our Polyner model over the EDS loss $\mathcal{L}_{\text{EDS}}$ on the DeepLesion [43] dataset.

| Parameter | PSNR |
| --- | --- |
| $\lambda = 0$ | 22.71±4.97 |
| $\lambda = 0.1$ | 33.75±2.30 |
| $\lambda = 0.2$ | **36.87±1.56** |
| $\lambda = 0.3$ | 36.83±1.52 |
| $\lambda = 0.4$ | 36.39±1.79 |

to the GT image. We present the quantitative results in Table 2. The results indicate that it contributes to an essential improvement of +3.92 dB in PSNR.

**Influence of Energy-dependent Smooth Loss.** We investigate the influence of the EDS loss on the model performance. To this end, we set different weights $\lambda \in \{0, 0.1, 0.2, 0.3, 0.4\}$ for the EDS loss in Eq. (9). We show the quantitative results in Table 3. The model performance initially improves with an increasing contribution of the EDS loss but later slightly degrades if its contribution keeps increasing. The best performance is obtained at $\lambda = 0.2$. This is because the absence of the EDS loss cannot ensure the smooth changes of the LACs of the body over energy levels. However, excessive bias towards it can result in the forward model degrading as a linear integral transformation, where the polychromatic LACs across all energy levels are identical. Fig. 5 shows the qualitative results. We observe that the CT images at different energy levels randomly change when the EDS loss is not used. In contrast, the regularization can effectively ensure the changes in the LACs are smooth. Moreover, the LACs slowly decrease as the X-ray energy level increases, which is generally consistent with the experimental findings reported in previous works [50].

**Influence of Number of Energy Levels.** We investigate how the number of discrete energy levels $N$ affects the model performance. We set the energy range to $[20, 120]$ and uniformly sample the $N = \{50, 76, 86, 101\}$ energy levels from the range. We also resample the normalized energy spectrum $\eta \in \mathbb{R}^N$ to match these levels. Table 4 shows the quantitative results, which indicate that increasing the energy resolution improves the MAR performance. We present the qualitative results in Fig. 6, which show that our Polyner with a higher energy resolution ($N = 101$) produces better CT results in terms of both image details and contrast. Our method approximates polychromatic X-rays with discrete energy levels, introducing an approximation error. Therefore, improving the energy's resolution may help to reduce this error and improve the model performance.

## 5   Conclusion & Limitation

We present Polyner, a novel method for the nonlinear MAR problem. The proposed Polyner follows the unsupervised learning paradigm and does not require any external training data, which greatly makes it useful in clinical scenarios. Our Polyner learns a neural representation of polychromatic

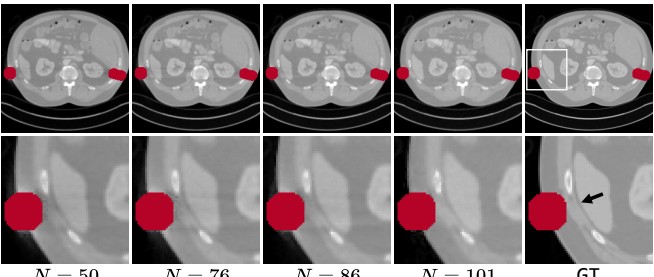

$N = 50$  $N = 76$  $N = 86$  $N = 101$  GT

Figure 6: Qualitative results of our Polyner ablating the discrete energy levels $N$ on a sample (#9) of the DeepLesion dataset [43]. The red regions denote metals.

Table 4: Quantitative results of our Polyner ablating the discrete energy levels $N$ on the DeepLesion [43] dataset.

| #Energy Levels | PSNR |
| --- | --- |
| $N = 50$ | 30.98±1.02 |
| $N = 76$ | 34.99±1.30 |
| $N = 86$ | 36.15±1.38 |
| $N = 101$ | **36.87±1.56** |

CT images to fundamentally avoid the nonlinear metal effect causing metal artifacts. It thus can reconstruct clean CT images from metal-affect measurements. Extensive experiments on three CT datasets demonstrate that our Polyner yields comparable or even better supervised DL methods.

Though the proposed Polyner achieves excellent MAR performance, there still are some limitations. Firstly, our Polyner is case-specific, meaning an independent model has to be optimized for each sample. Technically, the optimization of the Polyner for a CT image of 256×256 size requires about 2 minutes on a single NVIDIA RTX TITAN GPU (24 GB). In addition, our current Polyner model is based on 2D fan-beam and 3D cone-beam CT, while more advanced types of beams (*e.g.*, helical CT) are not implemented.

## 6 Acknowledgement

This work was supported by the National Natural Science Foundation of China under Grants No. 62071299 and MoE Key Lab of Intelligent Perception and Human-Machine Collaboration (ShanghaiTech University). Ce Wang was supported by the National Natural Science Foundation of China under Grants No. 62301532, in part by the Natural Science Foundation of Jiangsu Province under Grant No. BK20230282

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

# —Supplementary Material—
# Unsupervised Polychromatic Neural Representation
# for CT Metal Artifact Reduction

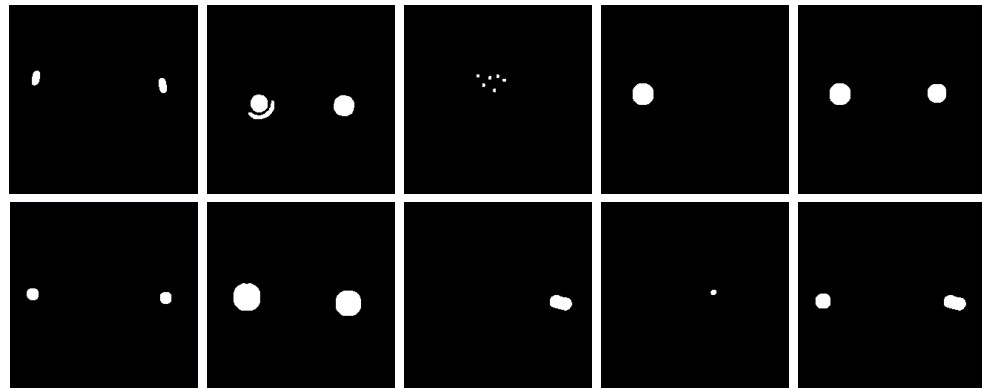

Figure 7: Ten shapes of metals for synthesizing metal-corrupted measurements in the DeepLesion [43] dataset. These metals are supposed as Titanium. The white regions denote metals.

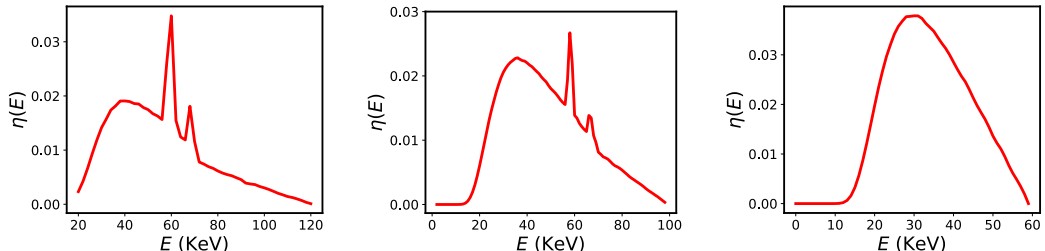

Figure 8: (*Left*) The simulated spectrum within an energy range of [20, 120] for synthesizing metal-corrupted measurements of the DeepLesion [43] and XCOM [44] datasets. The spectrums estimated within the energy ranges of [0, 100] and [0, 60] estimated by the SPEKTR toolkit [42] for the real walnut sample (*Middle*) and mouse thigh (*Right*).

## A  Additional Details of Datasets

In this work, we perform experiments on four datasets: DeepLesion [43], XCOM [44], and our Walnut Sample. DeepLesion and XCOM are simulation datasets, while Walnut Sample and Mouse Thigh are real-world datasets. *Note that our method is fully unsupervised, and thus all the data are exclusively used for testing purposes.*

**DeepLesion.**    To simulate metal-corrupted measurements in the DeepLesion dataset [43], we follow the pipelines outlined in [30, 7, 8]. For our experiments, we extract 200 2D images from the DeepLesion dataset as test GT samples [43]. As depicted in Fig. 7, we leverage ten shapes of metallic implants from [30, 7, 8] and consider them Titanium. To simulate the X-ray source, we employ a polychromatic X-ray with an energy range of [20, 120] KeV and a minimum energy unit of 1 KeV. The number of photons emitted by the X-ray source is set to $2 \times 10^7$. The normalized energy spectrum $\eta$ of the X-ray source is illustrated in Fig. 8 (*Left*). We adopt an equiangular fan-beam CT acquisition geometry, and the detailed parameters are provided in Table 5. Additionally, we incorporate Poisson noise and consider the partial volume effect in the sinogram domain during the simulation process.

**XCOM.**    For the XCOM dataset [44], we use two samples provided by Zhang *et al*. [7]. These two cases are simulated using two 2D clean CT images sourced from the XCOM [44] database. Zhang *et*

Table 5: Detailed parameters of the acquisition geometry for the four datasets.

| Parameters | Simulation Datasets | | Real Datasets | |
| --- | --- | --- | --- | --- |
| | DeepLesion [43] | XCOM [44] | Walnut | Mouse Thigh |
| Type of geometry | 2D fan-beam | 2D fan-beam | 2D fan-beam | 3D cone-beam |
| Source Voltage (kV) | 120 | 120 | 100 | 60 |
| Source Current (uA) | - | - | 200 | 200 |
| Exposure Time (ms) | - | - | 276 | 730 |
| Image size | 256×256 | 512×512 | 650×650 | 200×200×150 |
| Voxel size (mm) | 1×1 | 0.8×0.8 | 0.05×0.05 | 0.06×0.06×0.06 |
| Angle range (°) | [0, 360) | [0, 360) | [0, 360) | [0, 360) |
| The number of the angles | 360 | 984 | 720 | 900 |
| Angular spacing (°) | 0.1 | 0.057 | - | - |
| Detector spacing (mm) | - | - | 0.069 | 0.069 |
| Distance from source to center (mm) | 362 | 743 | 92.602 | 92.602 |
| Distance from center to detector (mm) | 362 | 743 | 65.946 | 65.946 |

*al.* [7] consider a polychromatic X-ray source with an energy range of [20, 120] KeV and a minimum energy unit of 1 KeV. The corresponding normalized energy spectrum $\eta$ is depicted in Fig. 8 (*Left*). In oder to generate metal-corrupted sinograms, an equiangular fan-beam CT acquisition geometry is employed, and the geometry parameters are specified in Table 5. Similar to the DeepLesion [43] dataset, Zhang *et al.* [7] also simulate Poisson noise and consider the partial volume effect in the sinogram domain during the simulation process.

**Walnut Sample.** To assess the performance of our proposed method on real CT data, we employ a commercial Bruker SKYSCAN 1276 micro-CT scanner to scan a walnut sample that contains a metal paper clip. Detailed parameters of the acquisition geometry can be found in Table 5. To estimate the X-ray spectrum of the micro-CT scanner, we leverage the SPEKTR toolkit [42]. The estimated spectrum is illustrated in Fig. 8 (*Middle*).

**Mouse Thigh.** We also scan a mouse thigh containing a metal intramedullary needle on the same micro-CT scanner. This sample is 3D cone-beam data. Detailed parameters of the acquisition geometry are shown in Table 5. We leverage the SPEKTR toolkit [42] to estimate the X-ray spectrum of the micro-CT scanner. The estimated spectrum is illustrated in Fig. 8 (*Right*).

# B  Additional Details of Baselines

In our experiments, we compare our proposed method against eight baseline MAR approaches. *Notably, for the five DL-based methods, we evaluate their performance using the pre-trained models provided by the respective authors.*

**FBP.** The FBP [3] is a conventional approach used for reconstructing CT images. It involves re-projecting the acquired sinogram data onto the image domain using the corresponding projection angles and geometric parameters to obtain an approximate estimate of the unknown image. In our experiments, we use the in-build function `ifanbem` in MATLAB (https://ww2.mathworks.cn/help/images/ref/ifanbeam.html?requestedDomain=cn).

**LI.** A simple strategy to mitigate metal artifacts in CT imaging involves the direct linear interpolation of the sinogram [10] to fill in the regions affected by metal. This approach does not require any network training but may result in imperfect sinogram completion, which in turn could introduce undesired artifacts in the reconstructed image. In our experiments, we use the implementation provided by Zhang *et al.* [7] (https://github.com/yanbozhang007/CNN-MAR/blob/master/cnnmar).

**NMAR.** The NMAR [11] introduces a generalized normalization technique that extends previously developed interpolation-based MAR techniques (*e.g.*, LI [10]), which also does not require any network training. Specifically, it normalizes the projections before interpolation based on forward projections of a prior image obtained through multi-threshold segmentation of the initial image. In our experiments, we use the implementation provided by Zhang *et al.* [7] (`https://github.com/yanbozhang007/CNN-MAR/tree/master/cnnmar`).

**CNN-MAR.** Zhang *et al.* [7] proposed a CNN-based MAR framework, which uses a CNN to estimate a prior image and subsequently apply a sinogram correction. However, despite the strong representation ability of CNNs, these approaches are still susceptible to secondary artifacts resulting from inconsistent sinograms. In our experiments, we use its official implementation and pre-trained model (`https://github.com/yanbozhang007/CNN-MAR/tree/master/cnnmar`).

**DICDNet.** Wang *et al.* [14] propose a deep interpretable convolutional dictionary network (DICD-Net) for the MAR task, which explicitly formulates the prior structures underlying metal artifacts in CT images as a convolutional dictionary model. In our experiments, we use its official implementation and pre-trained model (`https://github.com/hongwang01/DICDNet/tree/main`).

**ACDNet.** Similarly to DICDNet [14], the adaptive convolutional dictionary network (ACDNet) [13] explicitly encodes the prior observations underlying the MAR task into an adaptive convolutional dictionary network. In our experiments, we use its official implementation and pre-trained model (`https://github.com/hongwang01/ACDNet`).

**ADN.** The ADN [30] is an unsupervised learning approach for the MAR problem. Specifically, it takes unpaired metal-corrupted and clean CT images as inputs to learn the transformation between these two distributions. In our experiments, we use its official implementation and pre-trained model (`https://github.com/liaohaofu/adn/tree/master`).

**Score-MAR.** Song *et al.* [9] demonstrated that unconditional diffusion models can be adapted to various inverse problems, such as the MAR. Specifically, it learns the prior distribution of metal-free CT images with a generative model in order to infer the lost sinogram in the metal-affected regions. In our experiments, we use its official implementation and pre-trained model (`https://github.com/yang-song/score_inverse_problems`).

## C   Additional Visual Results

Figs. 9, 10, 11, 12, and 13 demonstrate some additional visual results. We observe that the proposed Polyner generally obtains the best MAR results.

## D   Broader Impacts

Our Polyner is expected to have significant broader impacts in the field of medical imaging. By effectively reducing metal artifacts in CT scans, our research has the potential to improve diagnostic accuracy, leading to more precise diagnoses and enhanced patient care. However, it is important to address potential limitations and concerns, such as the possibility of introducing false positive or negative results. Thorough evaluation and validation of our method is crucial to ensure its reliability and minimize any adverse effects. Overall, our work contributes to advancing medical imaging technology and has the potential to positively impact healthcare.

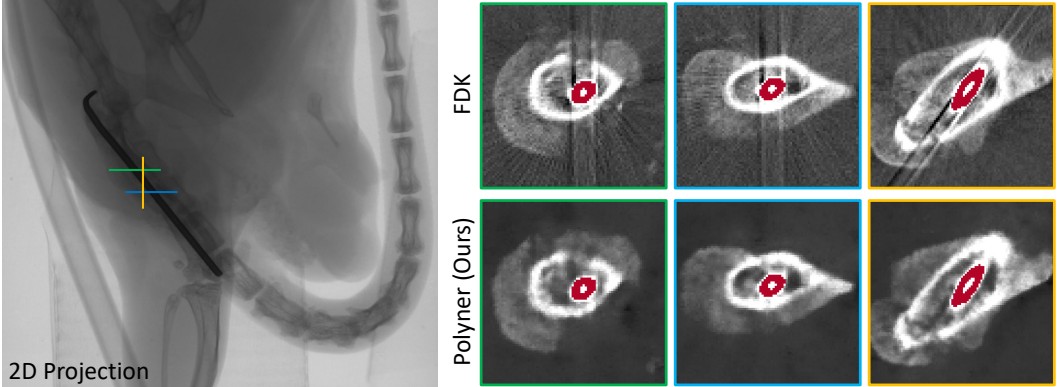

Figure 9: (*Left*) A sample among 2D projections of a mouse thigh containing a metal intramedullary needle scanned by the micro-CT scanner. (*Right*) Qualitative results of FDK [49] and our Polyner on the sample. Note that the acquisition geometry is the 3D cone beam. The reconstructed images have a size of 200×200×150. Our Polyner takes about 32 minutes on a single NVIDIA RTX TITAN GPU. The red regions denote the metal needle tubing. This data collection is approved ethically.

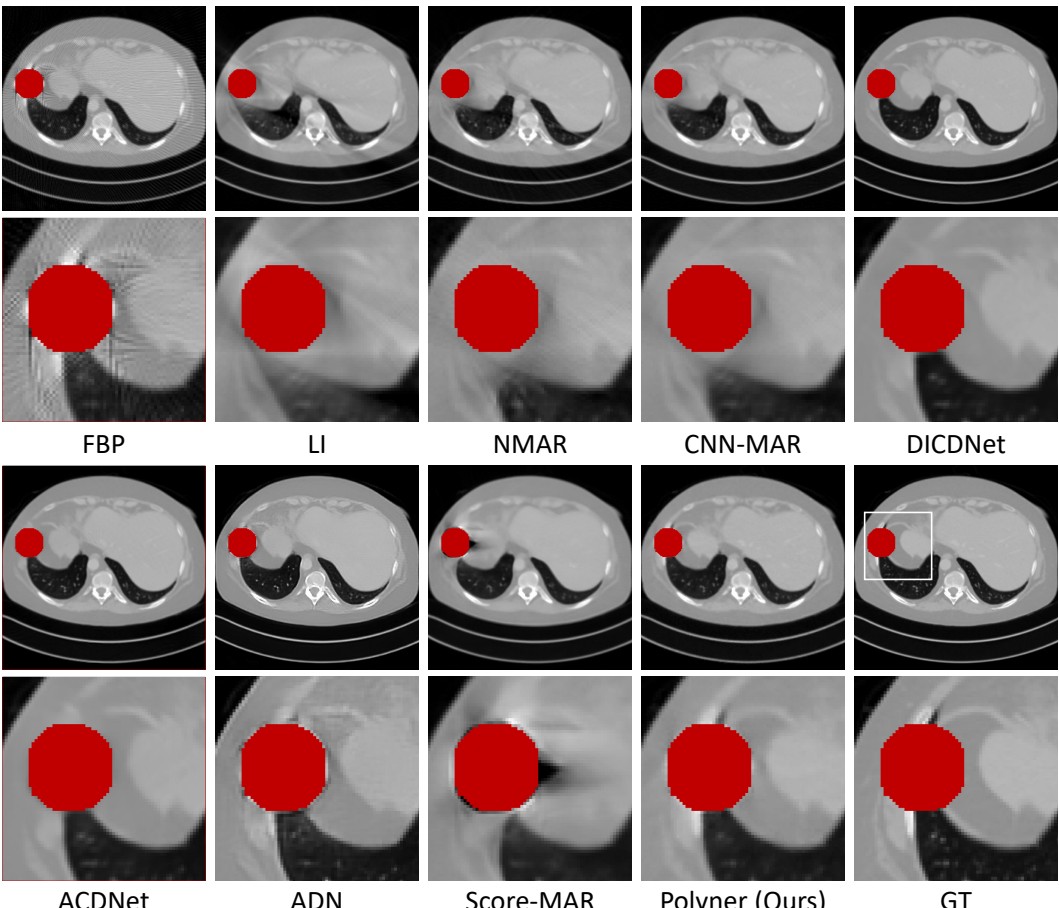

Figure 10: Qualitative results of the compared methods on a sample (#74) of DeepLesion [43] dataset. The white regions denote metals.

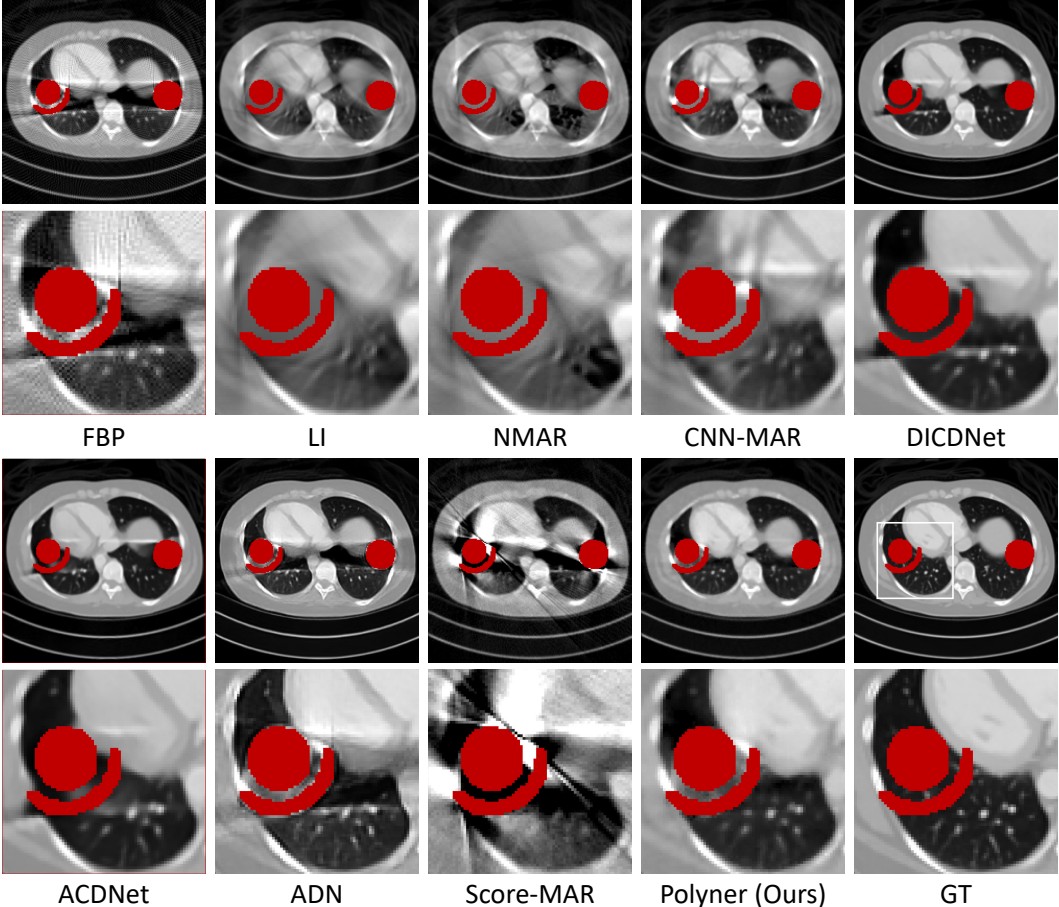

Figure 11: Qualitative results of the compared methods on a sample (#162) of DeepLesion [43] dataset. The white regions denote metals.

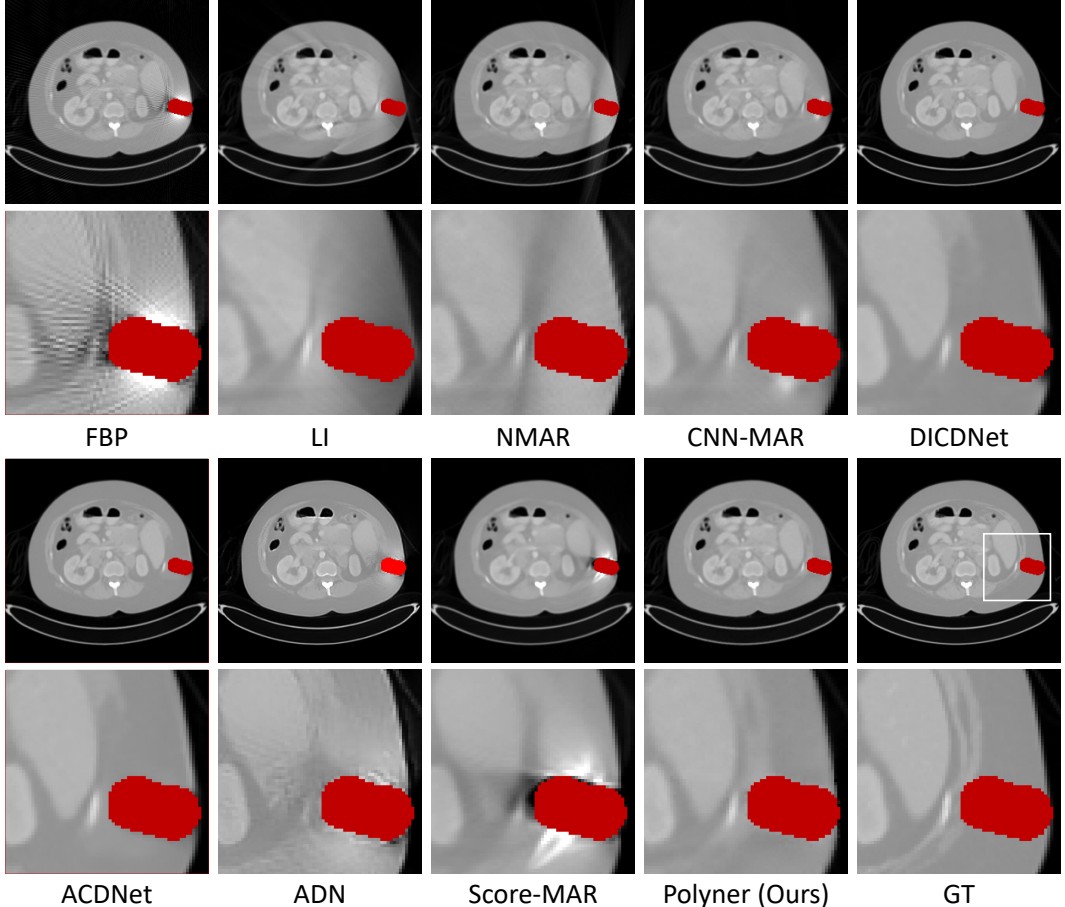

Figure 12: Qualitative results of the compared methods on a sample (#158) of DeepLesion [43] dataset. The white regions denote metals.

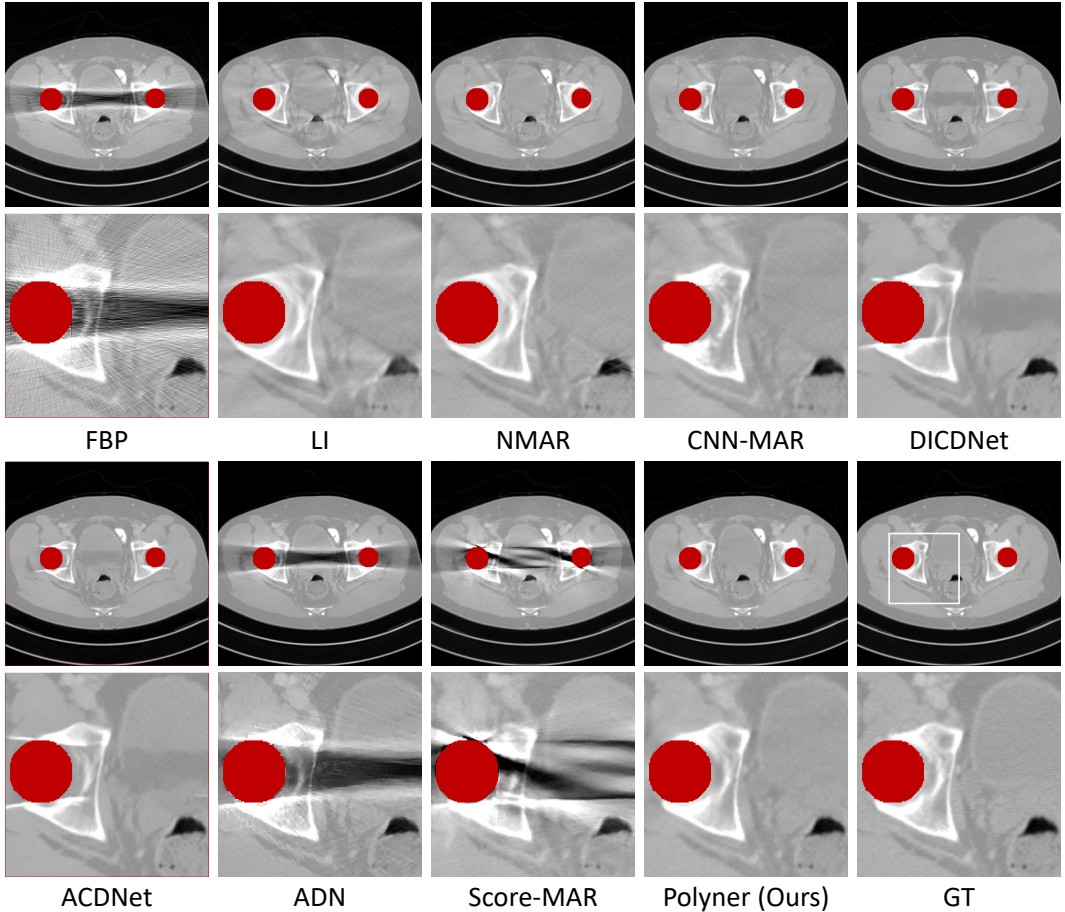

Figure 13: Qualitative results of the compared methods on a sample (#1) of XCOM [44] dataset. The white regions denote metals.

