# OpenReview forum: "Unsupervised Polychromatic Neural Representation for CT Metal Artifact Reduction"
_NeurIPS.cc/2023/Conference — NeurIPS 2023 poster_

### Official Review · Reviewer_QLvh · 2023-06-16

**Soundness:** 4 excellent
**Presentation:** 4 excellent
**Contribution:** 3 good
**Rating:** 8
**Confidence:** 4

**Summary:**

The authors tackle a very relevant problem in computed tomography: metal artifact reduction. They recognize some severe gaps in conventional methods (both model-based and deep-learning based) and propose a theoretically sound approach to solve these gaps using implicit neural representations in a somewhat similar fashion as, for example, NeRP. However, to solve the problem at hand, the first derive a polychromatic (as opposed to standard monochromatic) forward model to model the nonlinear CT acquisition process. In combination with a newly developed loss function, they use this model to apply some constraints to the physical properties of polychromatic CT and incorporate the implicit neural representation reconstruction approach. The method is not pre-trained and optimized per scan. They perform better than model-based and supervised deep learning methods on three datasets.

**Strengths:**

I must say that reading this paper was a pleasure. Overall, the manuscript is written very clearly and easy to follow. The ideas are original, and motivated well, and the results are discussed and explained appropriately.
I think the idea builds nicely upon previous work, aims to tackle a very relevant problem in medical imaging, and applies to a wide range of medical imaging settings. Therefore, the clinical relevance of the work is significant.
Section 2.1 provides a clear and sound theoretical foundation to understand their approach.
All ablation studies are supported by figures that show the importance of the components of the proposed architecture, and the other experiments are done on three different data sets.
It is a great strength that the method outperforms or performs equally compared to supervised models without using any training data.


**Weaknesses:**

I am quite familiar with INRs and methods such as NeRP, etc. I feel that the manuscript is complete and does not have severe weaknesses. There are, however, three things that I consider minor weaknesses. Firstly, the paper lacks an evaluation of a real-world patient data set. There are experiments on synthetically induced metal artifacts and the walnut scan. Still, in the end, the method should work on real patient data, and it would be better if the method were evaluated in that setting or if the authors would motivate why this was not done. Secondly, I think 2D fan beam CT is a bit old-fashioned. It would contribute to the strength of the paper – and its applicability in the real world – if the method would also work on cone-beam CT. Lastly, the speed of the approach limits the method significantly. 2 minutes for a 256x256 slice means about 4 minutes for a 512x512 slice. For brain CT, this would take 400 minutes, which is not clinically feasible. I suggest discussing potential methods to optimize the optimization process to make it more relevant for clinical application.

**Questions:**

1)	You assume that you can discretize the energy levels. Is this a valid assumption? Would a continuous approach be possible?
2)	Why have you not evaluated the method on a real-world patient data set?
3)	Would it be possible to extend the method to cone-beam CT or other CT acquisition protocols?
4)	How would you make the optimization faster to tackle the weakness regarding computational time?
5)	Why did you opt for ReLU and not SIREN activations, which are continuous, better differentiable, and have shown improved generalization capabilities?


**Limitations:**

The authors mention a sufficient amount of limitations of their work.

---

> ### Author Rebuttal · Authors · 2023-08-08
>
> We value the time and effort the reviewer dedicated to our work. It's genuinely uplifting to know that you consider our work "*reading this paper was a pleasure*". Below, we provide point-to-point responses to address your concerns.
>
> ---
>
> **Q1. You assume that you can discretize the energy levels. Is this a valid assumption? Would a continuous approach be possible?**
>
> **A1:** As expressed in Eq. 6, the estimated energy-dependent LAC maps are used to compute the projection data acquired with a polychromatic X-ray source. Theoretically, the energy of a polychromatic X-ray covers a continuous range. But in terms of the energy-dependent LAC maps, discretization is necessary due to the limitations of physical instruments and the trade-off between accuracy and computational costs in measurement. Hence, discretizing energy is a common and valid strategy in literature [1][2][3].
>
> Through our ablation study, it is shown that enhancing the resolution of the energy spectrum improves the MAR performance. To achieve a continuous energy spectrum, leveraging another INR network to learn its continuous representation could be a feasible solution. We would like to extend our method in future work.
>
> > [1] Punnoose, Jacob, et al. "spektr 3.0—A computational tool for x‐ray spectrum modeling and analysis." Medical physics (2016).
>
> > [2] Boone, John M., and J. Anthony Seibert. "An accurate method for computer‐generating tungsten anode x‐ray spectra from 30 to 140 kV." Medical physics (1997).
>
> > [3] Poludniowski, Gavin, et al. "SpekCalc: a program to calculate photon spectra from tungsten anode x-ray tubes." Physics in Medicine & Biology  (2009).
>
> ---
>
> **Q2. Why have you not evaluated the method on a real-world patient data set?**
>
> **A2:** There are two factors that hinder our ability to work with clinical patient data:
>
> - The raw projection data and geometry information usually are inaccessible on commercial clinical CT scanners.
> - Using real-world patient data could raise ethical concerns, and it is crucial to handle such data responsibly.
>
> In this preliminary work, our main objective is to present a novel unsupervised MAR method technically. In the future, we intend to conduct a comprehensive validation of our model using the clinical patient dataset, ensuring a thorough assessment of its performance and efficacy.
>
> Here we additionally scan a mouse thigh containing a metal needle tubing on a commercial Bruker SKYSCAN 1276 micro-CT scanner (Note this data collection is approved ethically). The qualitative results are shown in Figure R1 in the PDF file of the Global Response. Visually, our method still obtains a good reconstruction. We thus believe the efficiency of Polyner has been demonstrated by the walnut and this mouse data with inserted metal.
>
> ---
>
> **Q3. Would it be possible to extend the method to cone-beam CT or other CT acquisition protocols?**
>
> **A3:** See Q2 in the Global Responses, please.
>
> ---
>
> **Q4. How would you make the optimization faster to tackle the weakness regarding computational time?**
>
> **A4:** See Q1 in the Global Responses, please.
>
> ---
>
> **Q5. Why did you opt for ReLU and not SIREN activations, which are continuous, better differentiable, and have shown improved generalization capabilities?**
>
> **A5:** In our Polyner implementation, we leverage hash encoding to enhance the fitting capacity of the MLP network (the experiments about its effectiveness are provided in Q1 of the response to Reviewer ibgY). SIREN is also an effective INR architecture where a good continuous representation can be obtained.
>
> However, there are two main reasons we do not use SIREN activations in our Polyner:
>
> - An excellent performance can be achieved by the hash encoding with a simple MLP.
> - Using SIREN activation to replace ReLU will introduce an extra hyper-parameter $\omega_0$ (See [4] for more details). This might degrade the robustness of our model.
>
> Considering these factors, we use hash encoding and a simple MLP in our Polyner implementation, as it fulfills our requirements effectively. However, we also believe that exploring a more effective INR architecture is a straightforward and promising way to improve the MAR performance further.
>
> > [4] Sitzmann, Vincent, et al. "Implicit neural representations with periodic activation functions." NeurIPS. 2020.

---

> > ### Comment · Reviewer_QLvh · 2023-08-11
> >
> > I have read the other reviews and point-by-point responses from the authors.
> >
> > Global responses: Thanks for including an example of the 3D cone-beam reconstruction in the rebuttal. I understand that you prefer to mainly present the method in 2D, but I was wondering if the cone-beam results will be part of the paper or if you are going to leave them out?
> >
> > Q1) Ok, clear.
> >
> > Q2) I know that this info is usually not available, and that ethical concerns should be handled responsibly, but I don’t think it is that hard to acquire such data and handle it accordingly. Even one scan would help here. Will the mouse be included in the manuscript?
> >
> > Q3) Ok.
> >
> > Q4) Ok.
> >
> > Q5) Ok.

---

> > > ### Author Response · Authors · 2023-08-11
> > >
> > > Dear QLvh,
> > >
> > > Thanks for your prompt response!
> > >
> > > We fully agree with you that experimenting with real patient data can enhance the reliability of the model. We plan to validate the model comprehensively using clinical data in our future work. Additionally, we will incorporate the cone-beam reconstruction results for the mouse thigh in the revised submission.
> > >
> > > Best regards,
> > >
> > > Authors

---

### Official Review · Reviewer_oo65 · 2023-07-03

**Soundness:** 2 fair
**Presentation:** 4 excellent
**Contribution:** 2 fair
**Rating:** 4
**Confidence:** 5

**Summary:**

This paper introduces Polyner, an extension of implicit neural representation to a nonlinear inverse problem with a forward model that simulates the polychromatic nonlinear CT acquisition process. This design allows Polyner to reduce metal artifacts without external training data and exhibits better generalization to out-of-domain (OOD) data. Experimental results showcase improvements of Polyner on OOD datasets and clinical data while being competitive with state-of-the-art supervised methods.

**Strengths:**

1. The motivation is technically sound, and the proposed unsupervised method is interesting compared to the previous supervised paradigm. Taking into account the important knowledge of CT is reasonable.

2. The performance on OOD dataset and clinical images demonstrates the OOD capacity, though the quantitative evaluation (see weakness) is unfair. The ablation study confirms the effectiveness of the polychromatic CT forward model and the proposed loss function.

3. The paper is well-written and easy to follow. Experimental details are provided in supplementary materials with the open-sourced code.

**Weaknesses:**

1. While the author claims that "Polyner is superior to its supervised counterparts," this is not sufficiently supported by the experimental results and may be caused by unfair comparison.  Directly generalizing pre-trained models to a different size (e.g., 256x256) for testing might not yield optimal results due to variations in noise levels and geometry settings.  A fair comparison should re-train or fine-tune these methods to fit the specific geometry settings.

2. The suboptimal quantitative and visual results of ACDNet and DICDNet, in contrast to their original papers and CNNMAR, might be due to differences in geometry settings.  It is necessary to clarify whether the supervised models were fine-tuned on the IID setting of the DeepLesion dataset.  If not, please provide fine-tuned results (excluding OOD datasets) to ensure an accurate performance evaluation.

3. The color representation of CT images impedes readability.  Presenting grayscale results with an appropriate window to facilitate evaluation is more reasonable.  The goal of MAR is to remove artifacts **and** preserve diagnosis ability.

4. Please compare the proposed method with the state-of-the-art dual-domain methods published in 2022 or 2023.  Additionally, discuss the potential of the proposed method in enhancing traditional dual-domain methods.

5.  Since the proposed method is case-specific and requires optimization for each slice, it is essential to compare iterative reconstruction methods optimized in a similar manner. Moreover, reporting the computational time for a single scan will provide meaningful comparisons.

**Questions:**

1. It is important to clarify whether the DL methods (excluding CNNMAR) were fine-tuned on your dataset to match the geometry settings, following the training procedures described in their papers. If not, please provide the fine-tuned results for evaluation.

2. Please specify whether the PSNR and SSIM metrics were calculated based on specific windowing or attenuation coefficients.

3. Suggest changing the color scheme of the visual results. While it may be suitable for the ablation study to highlight details, grayscale images are more widely accepted in the field of CT imaging.

4. Sinogram inpainting-based methods should be reproduced and compared, which can show the advantages of the proposed polychromatic model.

5. Some statements are not accurate. (a) L36“..remove metal-affected extreme value signals....” Existing methods used different methods to replace these values. (b) L41 “collecting a large number of artifacts-free CT images...”, collecting such a dataset is easy. Instead, collecting a paired dataset is challenging. (c)  In the caption of Fig. 3, swap left and right.


**Limitations:**

In addition to these limitations in the paper, there is another limitation: When coming to 3D reconstruction, the proposed methods may produce discontinuous z-axis since it is a case-specific method for each slice.

---

> ### Author Rebuttal · Authors · 2023-08-08
>
> Thanks for your efforts and valuable comments. Below, we provide point-to-point responses to address your concerns.
>
> ---
>
> **Q1. Clarify whether the supervised ACDNet and DICDNet models were fine-tuned on the IID setting of the DeepLesion dataset**
>
> **A1:** The two supervised methods (ACDNet and DICDNet) and one self-supervised method (ADN) are pre-trained on the DeepLesion dataset. However, the size of CT images for the first two is 416×416, but 256×256 for the last one. In our experiments, we uniformly test them on the DeepLesion dataset consisting of images of 256×256 size, called as DeepLesion (256), without conducting any fine-tuning. We acknowledge that this might slightly degrade the performance of ACDNet and DICDNet, weakling our claim “*Polyner is superior to its supervised counterparts*.”
>
> To address this concern, we construct another DeepLesion dataset consisting of images of 416×416 size, called as DeepLesion (416), *fully following the geometry settings reported in the original papers of ACDNet and DICDNet*. Then, we compare them with our Polyner on the new DeepLesion (416). Note that all the hyper-parameters for optimizing our Polyner are the same as in our original submission, which demonstrates its robustness.
>
> We show the quantitative results in Table R3. There are three observations:
> - The performances of ACDNet and DICDNet are similar to their original reports on the new DeepLesion (416). For example, ACDNet achieves 40.68 dB and 40.91 dB of PSNR in the original paper and DeepLesion (416), respectively. We thus hold the performance concern caused by geometry settings is resolved.
> - Compared with the DeepLesion (256), the three methods all achieve higher performance on the DeepLesion (416). The reason is the number of projections is 360 and 640 for the DeepLesion datasets (256) and (416), respectively. Increasing the projections can improve reconstruction accuracy.
> - Our method achieves the best reconstruction performance quantitatively. For instance, PSNR improves by 1.04 dB and 1.27 dB compared with ACDNet and DICDNet, respectively.
>
> We show the qualitative results in Figure R3 in the PDF file of the Global Response. The result of our Polyner is closest to the GT sample. This experiment shows our Polyner can perform compared to the supervised methods for the IID settings of in-domain datasets.
>
> We will add this experiment for geometry settings into the revised manuscript for a more comprehensive evaluation.
> |	|DeepLesion (256)|DeepLesion (256)|	DeepLesion (416)|DeepLesion (416)|
> | :---: |:---: |:---: |:---: |:---: |
> ||PSNR|SSIM|PSNR|SSIM|
> |DICDNet|	37.55±2.52|0.9689±0.0116|	40.68±1.79|0.9786±0.0061|
> |ACDNet|	**38.19±2.54**|0.9675±0.0152|	40.91±2.23|0.9753±0.0120|
> |Polyner (Ours)|	37.57±1.93|**0.9754±0.0083**|	**41.95±1.68**|**0.9829±0.0052**|
>
> *Table R3: Quantitative results of DICDNet, ACDNet, and our Polyer on the two versions of DeepLesion datasets.*
>
> ---
>
> **Q2. How to compute PSNR and SSIM metrics?**
>
> **A2:** The two metrics are calculated based on the attenuation coefficients.
>
> ---
>
> **Q3. Suggest changing the color scheme of the visual results.**
>
> **A3:** In the original submission, we use a color scheme to prominently visualize the differences among reconstructed images. However, we fully agree with Reviewer oo65 that the grayscale scheme is more standard and widely used in CT imaging. To enhance readability, we will replace the color images with a grayscale version in the revised manuscript.
>
> ---
>
> **Q4. Supervised dual-domain methods should be compared.**
>
> **A4:** We choose not to include dual-domain MAR methods for two primary reasons:
> - Based on the experimental results reported by the original papers, the SOTA methods ACDNet and DICDNet outperform the well-known dual-domain MAR methods, such as DuDoNet and DuDoNet++.
> - The official code and pre-trained models of the majority of dual-domain methods are not available, making a reliable comparison challenging.
>
> Taking these factors into account, we have decided to use ACDNet and DICDNet as the main deep learning-based comparison methods for our study.
>
> ---
>
> **Q5. Some statements are not accurate.**
>
> **A5:** We appreciate the reviewer’s patient proofreading. We will carefully modify these inaccurate statements and typos in the revised submission.
>
> ---
>
> **Q6. For 3D reconstruction, Polyner may produce discontinuous z-axis.**
>
> **A6:** As shown in Figure R1 in the PDF file of the Global Responses, we have developed a 3D cone-beam version of Polyner and successfully used it to reconstruct a 3D image of a mouse. The extension of Polyner to support 3D CT imaging is straightforward by modifying the scanning geometry used for simulating the X-rays, and the superior MAR performance is stable. The image continuity along the z-axis is improved in the 3D implementation result.
>
> We present Polyner in the 2D version to ease a broader comparison with existing MAR works. Due to the training efficiency and memory footprint considerations, most deep-learning MAR models are 2D. A 3D version could limit the fairness of our comparison.
>
> ---
>
> **Q7. Comparison with iterative methods and computational time.**
>
> **A7:** Thank you for the suggestion. We have added the algebraic reconstruction technique (ART), a classical iterative CT imaging method, as additional baselines. Table R4 displays the quantitative results. Two observations are evident:
>
> - As iterative methods, both ART and our Polyner are markedly slower than the analytical FBP algorithm.
> - Our Polyner achieves significant improvements compared to the two conventional methods.
>
> We will include this comparison in the revised manuscript.
>
> ||PSNR|SSIM|Average Time|
> | :---:|:---:|:---:|:---:|
> |FBP	|29.17±3.30|	0.7231±0.0998|**0.06 s**|
> |ART	|32.88±3.63|	0.8352±0.0701|180 s|
> |Poylner (Ours)|**37.57±1.93**|**0.9754±0.0083**|121 s|
>
> *Table R4. Quantitative results of FBP, ART, and our Polyner on the DeepLesion dataset. Note that ART and Polyner are all iterative methods.*

---

> > ### Comment · Reviewer_oo65 · 2023-08-16
> >
> > Thank the authors for their additional results and detailed responses. However, the following concerns remain unsolved.
> >
> > 1. The authors clarified that "the two metrics are calculated based on the attenuation coefficients." A typical way is to select a specific CT window for metric calculation and comparison. For example, in Dudonet++, the metrics are evaluated in a window of [-175, +275] HU. A larger window leads to a higher PSNR value. This work used attenuation coefficients for metrics calculation, corresponding to the largest CT window.
> >
> > 2. This work used the trained models from the original literature for comparison. However, these models may use different training sets (patients), simulation parameters, preprocessing, and CT windows for training. Such factors could cause unfairness. Referring to both Figures 2 and 3 in the global response, it is evident that both ACDNet and DICDNet introduce noticeable artifacts, especially adjacent to the left metal. Such effects should not be present in a well-trained model. Given the source codes and checkpoints, why not re-train the model and compare the trained models with the checkpoints provided by the authors?
> >
> > 3. The proposed method requires 2 mins to reconstruct one slice, which may be far from clinical applications. One scan (>100 slices) should be done in less than 1 min.
> >
> > In summary, the present work is interesting, in line with other reviewers. I believe a fair experimental comparison could make this work much better. Therefore, I would keep the previous rating but don't mind this paper being accepted.

---

> > > ### Author Response · Authors · 2023-08-16
> > >
> > > Dear oo65,
> > >
> > > Thank you for your reply and additional suggestions！
> > >
> > > Below are our point-by-point responses to your concerns.
> > >
> > > ---
> > >
> > > **A1:** When using an HU window to compute PSNR, the differences among the intensities beyond the window are not being considered. In DuDoNet++ [1], a window of [-175, +275] HU is selected to emphasize the intensity differences among soft tissues and bone (e.g., fat [-100, -90]; liver [50, 70]; kidney [20-40]; cancellous bone [50, 200]). However, it ignores the HU levels for air (-1000 ± 10), tooth enamel [800, 1200], and compact bone [250, 2500]. In our test data, we compare CT MAR reconstruction in abdominal, dental, lung, and brain imaging, where the HU value of tooth enamel, air (intestinal lumen), pulmonary air, skull, and rib all are beyond this range. Therefore, we compute the PSNR metric on the raw attenuation coefficients. We hold the fairness for our comparison.
> > >
> > > Here, we also fully follow the scheme reported in DuDoNet++ [1] to compute the PSNR metric. Table R6 presents the quantitative results. The performance of all methods slightly decreases but remains satisfactory. Moreover, our Polyner produces comparable performance to the supervised ACDNet and DICDNet, which is consistent with our current evaluation.
> > >
> > > | |DeepLesion (256)-AC|	DeepLesion (256)-HU|	DeepLesion (416)-AC|	DeepLesion (416)-HU|
> > > |:---:|:---:|:---:|:---:|:---:|
> > > |DICDNet	|37.55±2.52	|36.01±2.07|	40.68±1.79	|37.55±1.84|
> > > |ACDNet|	**38.19±2.54**	|**37.00±2.13**|	40.91±2.23	|37.94±2.57|
> > > |Polyner (Ours)|	37.57±1.93|	36.25±1.88|	**41.95±1.68**|	**38.68±1.32**|
> > >
> > > *Table R6: Quantitative results of DICDNet, ACDNet, and our Polyner on the two versions of the DeepLesion datasets by using two computational strategies. AC denotes PSNR metrics based on the raw attenuation coefficients, while HU represents PSNR metrics based on a window of [-175, +275] HU, which is consistent with the approach used in DuDoNet++ [1].*
> > >
> > > > [1] Lyu, Yuanyuan, et al. "Dudonet++: Encoding mask projection to reduce ct metal artifacts." arXiv preprint arXiv:2001.00340 (2020).
> > >
> > > ---
> > >
> > > **A2:** The supervised ACDNet and DICDNet models are trained on the DeepLesion dataset. In Table R3, we present their quantitative results on the two versions of the DeepLesion datasets (Please note that DeepLesion (416) is constructed by fully following the geometry settings in the original papers). Our obtained quantitative performance of ACDNet and DICDNet closely aligns with the results reported in the original papers. Therefore, we believe a fair comparison is guaranteed.
> > >
> > > Furthermore, the results of ACDNet and DICDNet exhibit noticeable artifacts. The primary reason could be the diversity of the test samples. The MAR is a very challenging problem, and since ACDNet and DICDNet are supervised methods, it is common for such methods to experience drops in performance due to challenges in generalization.
> > >
> > > Finally, we maintain that utilizing checkpoints provided by the authors can ensure a relatively fair comparison, as training supervised models from scratch necessitates the selection of numerous hyperparameters. Consequently, we assess these baselines using the pre-trained checkpoints.
> > >
> > > ---
> > >
> > > **A3:** We agree with Reviewer oo65 that optimization speed is a limitation of our current Polyner. However, we believe we are discussing more advanced trends for solving the MAR task rather than providing a direct clinical solution. When needed, the optimization speed can be significantly improved by combining metal-learning [2] to learn a fine initialization (For more details, please refer to Q1 of the Global Response).
> > >
> > > > [2] Tancik, Matthew, et al. "Learned initializations for optimizing coordinate-based neural representations." CVPR. 2021.
> > >
> > > ---
> > >
> > > We hope our responses above can solve your concerns. If you have any additional questions, we eagerly look forward to engaging in further discussions.
> > >
> > > Best regards,
> > >
> > > Authors

---

### Official Review · Reviewer_KPnC · 2023-07-06

**Soundness:** 3 good
**Presentation:** 2 fair
**Contribution:** 2 fair
**Rating:** 5
**Confidence:** 4

**Summary:**

The paper with title: Unsupervised Polychromatic Neural Representation for CT Metal Artifact Reduction presents an Implicit neural representation-based method for CT metal artifacts reduction, outperforming existing supervised and unsupervised approaches.


**Strengths:**

1. This paper presents a novel INR-based method for CT metal artifacts reduction.
2. The authors present a non-linear forward model to model the metal artifacts, and leverages it as signal domain loss function.
3. I appreciate the real-scan results which demonstrate the results on prospective corrupted dataset.

**Weaknesses:**

1. Regarding the results for the real-data - I noticed that for real-data, the gap or improvements of Polyner is not as significant as the simulated dataset, ACDNet is also doing a decent job, could you elaborate on this?
2. Regarding the forward model, from the ablation studies, including the non-linear components contribute to the improved performance, however, the forward model is still not perfect without considering some system attributes. Can you elaborate on this aspect?
3. lack of quantitative results on real-data.

**Questions:**

I have put my questions in the weakness section.

**Limitations:**

1. The inference time is long if given limited performance improvements.

---

> ### Author Rebuttal · Authors · 2023-08-06
>
> Thank you for taking the time to review our work. We are pleased to receive your positive feedback. Below, we provide point-to-point responses to address your concerns.
>
> ---
>
> **Q1. Regarding the results for the real-data - I noticed that for real-data, the gap or improvements of Polyner is not as significant as the simulated dataset, ACDNet is also doing a decent job, could you elaborate on this?**
>
> **A1:** For MAR methods, the performance is significantly influenced by the volume and shape of the metallic implant. In our real-data scanning, the metallic object exhibits a compact structure with a volume much smaller than those in the simulation data. The metal streaking artifacts are narrow and are not as complicated as that in the simulation data, as shown in the reconstruction result by FBP in Figure 3, while ACDNet takes the images by FBP as one of the inputs. Therefore, the reconstruction results of ACDNet are also acceptable.
>
> However, residual streaking artifacts are still observed by the results of ACDNet, and the signal intensities obviously deviate from the reference image. In contrast, our Polyner provides high-fidelity reconstruction with neglectable streaking artifacts.
>
> ---
>
> **Q2. Regarding the forward model, from the ablation studies, including the non-linear components contribute to the improved performance, however, the forward model is still not perfect without considering some system attributes. Can you elaborate on this aspect?**
>
> **A2:**  CT acquisition is a complicated process with various applications. Polyner is primarily proposed for medical CT, the most widely beneficial application among all application scenarios. In medical CT, commercial scanners adhere to guidelines for biological CT scans, characterized by controlled conditions: normal-dose X-rays, full projection views, exposure time, standard SNR, well-calibrated energy sources, and signal sensors. In such a setup, we propose our forward model primarily focuses on simulating the nonlinear effects encountered when scanning a human body with metal implants. We hold that in a relatively stable environment, the nonlinear forward model covers the most critical contributors in MAR CT reconstruction. The descent performance on real data also proves the efficiency of the forward model.
>
> Moreover, our Polyner model can easily be extended to tackle more complicated CT scanning situations. For instance, we could incorporate explicit regularization terms such as total variation into the loss function (Eq. 9) to compensate for the low SNR data acquisition in low dose CT condition [1]; we could combine a reprojection strategy for sparse view CT condition [2]; we could jointly optimize the INR network and a transformation matrix when the scanner sensor calibration is not perfect to correct motion between each projection [3].
>
> > [1] Zang, Guangming, et al. "IntraTomo: self-supervised learning-based tomography via sinogram synthesis and prediction." CVPR. 2021.
>
> > [2] Wu, Qing, et al. "Self-supervised coordinate projection network for sparse-view computed tomography." IEEE TCI (2023).
>
> > [3] Wang, Zirui, et al. "NeRF--: Neural radiance fields without known camera parameters." arXiv preprint arXiv:2102.07064 (2021).
>
> ---
>
> **Q3. Lack of quantitative results on real-data.**
>
> **A3:** In our real-data experiment, we only provide qualitative results. The main reason is that the reference CT image cannot be completely considered a ground truth due to non-rigid deformation caused by the insertion of the metal paper clip. This misalignment will affect the accuracy of quantitative metrics. Here, we calculate the quantitative results in Table R2. Our Polyner still performs the best. We will add these quantitative results to the supplementary material in the revised manuscript.
>
> |	|FBP|	LI|	ACDNet|	DIDCNet|	Polyner (Ours)|
> | :---: |:---: |:---: |:---: |:---: |:---: |
> |PSNR|	20.66|	21.94|	21.53|	21.61|	**22.00**|
> |SSIM|	0.3623|	0.4853|	0.3592|	0.3562|	**0.5484**|
>
> *Table R2: Quantitative results of the compared methods on the real collected walnut sample.*
>
> ---
>
> **Q4. The inference time is long if given limited performance improvements.**
>
> **A4:** See Q1 in the Global Response, please. And we would like to emphasize our Polyner contributes the most in proposing a fully unsupervised MAR method. At the same time, the performance is superior compared with existing supervised MAR methods.

---

> > ### Comment · Reviewer_KPnC · 2023-08-11
> >
> > I appreciate the authors for the thoughtful responses, I looked through other reviews and it answers a large portion of my concerns.
> > Regarding Q1: you mentioned:  In our real-data scanning, the metallic object exhibits a compact structure with a volume much smaller than those in the simulation data. Just wanna check if I understood correctly, you were saying since the metal is smaller, where the improvements are less significant. I wonder if you simulate with a smaller object, what will be the results? Or you perform the real-scan with a larger metal. Please follow up with this question.
> >
> > Regarding Q2: That's fair, thanks for your response!
> >
> > Regarding Q3: Sounds good.

---

> > > ### Author Response · Authors · 2023-08-12
> > >
> > > Dear KPnC,
> > >
> > > Thanks for your prompt reply!
> > >
> > > The fact aligns exactly with your understanding, i.e., the improvements by our model are more significant for larger-sized metals.
> > >
> > > In our experiments on the simulation DeepLesion, we evaluate model performance on ten different sizes of metals. The original submission reports average results for these sizes. Here, we additionally show three different sizes of metals in Table R5. The results reveal that our Polyner exhibits more pronounced advantages for larger-sized metals.  We will add the related discussion in the revised submission.
> > >
> > > |Metal Size|#Metal's Pixels|ACDNet|Polyner (Ours)|Improvements|
> > > |:---:|:---:|:---:|:---:|:---:|
> > > |Small|197|42.73 dB|42.47 dB|-0.26 dB|
> > > |Medium|1160|	41.84 dB|	42.00 dB|	+0.16 dB|
> > > |Large|3260|38.31 dB|40.78 dB|+2.47 dB|
> > >
> > > *Table R5: Quantitative results of ACDNet and our Poylner for three different sizes of metals on the DeepLesion (416) dataset.*
> > >
> > > Best regards,
> > >
> > > Authors

---

### Official Review · Reviewer_ibgY · 2023-07-06

**Soundness:** 4 excellent
**Presentation:** 4 excellent
**Contribution:** 4 excellent
**Rating:** 7
**Confidence:** 4

**Summary:**

The paper proposes a new implicit neural representation (INR)-based polychromatic x-ray CT reconstruction technique called Polyner. In normal CT reconstruction, the tissues of the body do not vary substantially in their attenuation coefficients, so the overall forward operator can be simply modeled as linear. This changes in the presence of metal implants, because the attenuation of metal implants depends on the energy of the input X-rays, giving a polychromatic attenuation response. Several methods have been proposed for polychromatic CT reconstruction, but these methods often manifest as inpainting methods that suffer from out-of-distribution generalization issues.

The present paper addresses this issue by using a NeRF-like model for estimating the attenuation coefficients in the body. The model is optimized by passing the predicted attenuations through a forward integral model and minimizing the distortion vs. the measurements. The forward model is adapted for metal vs. non-metal regions via mask.

Results presented in the paper are promising: the proposed Polyner method is competitive with past methods on in-domain test datasets in terms of PSNR and SSIM metrics on the DeepLesion and XCOM datasets. Competing methods have wide variances in performance among the two datasets, whereas Polyner is more stable. On a new CT dataset consisting of a walnut scan with a metal paperclip, where Polyner further establishes is strong robustness. In ablations, the paper establishes the importance of decisions taken with respect to the forward model and loss functions.

**Strengths:**

**Originality**
- The paper presents a novel approach for the metal artifact reduction problem. Although implicit neural representations have been used for previous CT reconstruction tasks, the metal artifact reduction problem is substantially different in terms of the physics under consideration and the artifacts that can manifest.

**Quality**
- The paper includes all the expected experiments, examining multiple simulation datasets as well as new real-world data, providing strong experimental validation of the proposed method.
- Comparison methods seem up to date, including methods as recent as 2022.

**Clarity**
- The paper is clear in its motivations for the metal artifact reduction problem and the justification for the INR approach.

**Significance**
- Providing a robust reconstruction method for CT with metal artifacts is a long-standing goal of the CT field, and it seems that this paper makes a god contribution towards this goal.

**Weaknesses:**

**Originality**
- The presence of previous INR-style methods in CT arguably reduces the originality of the current paper, but in my opinion not in a significant manner.

**Questions:**

1. Most ablations consider loss functions and forward models - did the authors consider modifications to the INR architecture, or is this not an area with large expected benefits?
2. Please specify more details on the binary mask - it was not clear to me how this is calculated or used from the text.

**Limitations:**

Limitations are listed with the discussion. I normally prefer a separate listing, but I do not have a major objection to this presentation.

---

> ### Author Rebuttal · Authors · 2023-08-06
>
> Thank the reviewer for the valuable comments. We are encouraged by your recognition of our work. Below, we provide point-to-point responses to address your concerns.
>
> ---
>
> **Q1. Most ablations consider loss functions and forward models - did the authors consider modifications to the INR architecture, or is this not an area with large expected benefits?**
>
> **A1.** The INR architectures usually consist of an encoding module and an MLP network. The encoding module transforms low-dimensional coordinates into high-dimensional embeddings, significantly enhancing the fitting capacity of the subsequent MLP network. In our Polyner model, we employ the SOTA hash encoding for embedding, which has become a standard module in many INR-based approaches [1]. Here, we compare it with position encoding [2] and Fourier encoding [3] on the DeepLesion dataset. From Table R1, we can see that hash encoding achieves the best performance in terms of both reconstruction quality and speed. Qualitative results are shown in Figure R2 of the PDF file of the Global Response.
>
> We will include this ablation study for the INR architecture in the supplementary material.
>
> In summary, the INR architectures do impact the model performance, particularly in terms of optimization speed. Hence, adopting a more powerful architecture can significantly improve the proposed Polyner model.
>
> |	|Position Encoding|	Fourier Encoding|	Hash Encoding|
> | :---:| :---:| :---:| :---:|
> |PSNR	|34.55±1.64|	35.07±1.42|	**36.87±1.56**|
> |Average Time	|10 min.|	8 min.|	**2 min.**|
>
> *Table R1: Qualitative results of our Polyner implemented by three different INR architectures on the DeepLesion dataset.*
>
> > [1] Tewari, Ayush, et al. "Advances in neural rendering." Computer Graphics Forum. 2022.
>
> > [2] Mildenhall, B., et al. "Nerf: Representing scenes as neural radiance fields for view synthesis." ECCV. 2020.
>
> > [3] Tancik, Matthew, et al. "Fourier features let networks learn high frequency functions in low dimensional domains." NeurIPS. 2020.
> ---
>
> **Q2. Please specify more details on the binary mask - it was not clear to me how this is calculated or used from the text.**
>
> **A2.** Due to the substantial distinctions in the linear attenuation coefficients (LACs) between metallic substances and biological tissue, we applied a straightforward threshold segmentation to the CT images reconstructed by FBP to generate these binary metal masks.
>
> In the EDS regularization term (Eq. 8), we use these masks to distinguish between metal and body regions and thus can enforce a smooth prior along the energy spectrum for the body area. We will include these details in the revised manuscript.

---

> > ### Comment · Reviewer_ibgY · 2023-08-15
> >
> > I would like to thank the authors for their reply. All of my questions are addressed. I have also read the other reviews. I thought there were some good comments by Reviewer oo65, but these were well-addressed by the authors in their rebuttal. I continue to hold my rating of 7.

---

> > > ### Author Response · Authors · 2023-08-15
> > > **Thank you for your efforts**
> > >
> > > Dear ibgY,
> > >
> > > Thank you for your prompt response!
> > >
> > > We are delighted to learn that your concerns have been addressed. Your constructive comments have truly enhanced our work. Once again, thank you for your efforts!
> > >
> > > Best regards,
> > >
> > > Authors

---

### Author Rebuttal · Authors · 2023-08-04

### **Global Responses**

We sincerely thank the reviewers for their insightful comments and suggestions!

We are encouraged by the reviewers' recognition of the novelty (ibgY, KPnC, QLvh), strong motivation (ibgY, oo65, QLvh), technical (oo65, QLvh) and theoretical (QLvh) soundness of our research. Their positive remarks on the anticipated experiments (ibgY, KPnC, QLvh), high performance (ibgY, KPnC, oo65, QLvh), and the clarity of our writing (ibgY, oo65, QLvh) are also highly appreciated.

Here we address two common concerns raised by multiple reviewers. Subsequently, we provide detailed responses to the individual feedback provided by each reviewer.

---

**Q1. Acceleration of optimization process (KPnC, oo65, QLvh)**

**A1:** Polyner uses implicit neural representation (INR) as its core architecture, and in light of the recent advancements in INR, there have been a few methods to speed up image reconstruction during the last two years. In our work, the current Polyner takes about 2 minutes to reconstruct a 2D CT image of 256×256 on a single NVIDIA RTX TITAN GPU.

There are three possible solutions to further accelerate the reconstruction:
- *Enhanced encoding schemes*. While the hash encoding used in our Polyner implementation currently represents the state-of-the-art (SOTA), exploring more powerful encoding schemes is a straightforward and promising approach.
- *Meta-learning*. An alternative solution involves using meta-learning to facilitate a superior initialization for our Polyner model. Empirically, Tancik et al. [1] demonstrated that meta-learning can significantly accelerate the optimization of INR networks across diverse tasks, including image fitting, sparse-view CT reconstruction, and novel view synthesis. However, this meta-learning-based solution requires an external dataset consisting of numerous artifact-free CT images, which might limit its applicability in certain rare cases.
- *Specialized acceleration chips*. The proposed Polyner is a deep learning-based method and is training based on general-purpose GPUs. Nevertheless, we could use some specialized acceleration chips to significantly acceleration the optimization.

> [1] Tancik, Matthew, et al. "Learned initializations for optimizing coordinate-based neural representations." CVPR. 2021.

---

**Q2. Extension of our Polyner model to 3D CT imaging (oo65, QLvh)**

**A2:** Our current Polyner model is presented based on 2D fan beam CT. And it can be extended easily for advanced CT acquisition protocols. For instance, in the case of 3D cone-beam settings, only a simple modification following 3D acquisition geometry for X-ray simulation needs to be conducted, while all other steps remain unchanged. One benefit of 3D reconstruction could be the local consistency along the z-axis in the reconstructed image. In addition, there is a potential problem in 3D CT images: The INR network requires a more powerful capacity to represent 3D objects, especially for large image dimensions, which may intensify computational costs, such as memory footprint.

As shown in Figure R1 in the PDF file of the Global Responses, we implement a 3D cone-beam version of the Polyner on a mouse CT scan conducted with a commercial Bruker SKYSCAN 1276 micro-CT scanner (Note this data collection is approved ethically). Our 3D Polyner model was employed to reconstruct a mouse thigh region with a size of 200×200×150. The enlarged figures in Figure R1 illustrate the mouse thigh regions containing metal needle tubing. The whole optimization process requires approximately 10 GB of memory and takes around 32 minutes when executed on a single NVIDIA RTX TITAN GPU. Visually, our method still obtains a good reconstruction, as shown in Figure R1.

Besides, we prefer to present Polyner in the current 2D fan beam version to facilitate a broader comparison with the recent deep learning-based MAR methods. Due to the training efficiency, most deep-learning MAR models are 2D. A 3D version could limit the fairness of our comparison.

---

> ### Author Response · Authors · 2023-08-21
> **Scheduled updates in the revised submission**
>
> Dear AC and all reviewers,
>
> We really enjoy the discussion with you, which has truly enhanced our work.
>
> We will make the following changes in the revised submission according to your suggestions, including
>
> - Add the 3D cone-beam reconstruction results of the mouse tight (Figure R1 in PDF of the Global Response) [by QLvh].
> - Add the ablation study for INR architectures (Table R1) [by ibgY and QLvh].
> - Add the quantitative results for the real-world walnut sample (Table R2) [by KPnC].
> - Add the comparison on the new DeepLesion (416) dataset (Table R3) [by oo65].
> - Add ART, a traditional iterative reconstruction method, as an additional baseline (Table R4) [by oo65].
> - Replace the color representation of qualitative results with a grayscale version [by oo65].
> - Correct all typos and inaccurate statements [by oo65].
>
> Thanks again for your efforts in reviewing our work!
>
> Best regards,
>
> Authors

---

### Decision · Program_Chairs · 2023-09-21

**Decision:**

Accept (poster)

**Comment:**

The final score of this paper is SA, A, BA ,BR. Although one reviewer has raised concerns about experimental comparisons, other reviewers found that the proposed method novel and has made sufficient technical contributions. The rebuttal has addressed most of the concerns except for the real-world patient data experiments. The AC has checked the paper, the review comments and the rebuttals, and believe that the paper has made sufficient contributions for the acceptance of NeurIPS. The AC strongly encourages the authors to finish the real-world patient data experiments and incorporate it into the final version of the paper. This acceptance is also based on the condition that the authors promised to released the source codes of this work.